# Ctrl-DNA: Controllable Cell-Type-Specific Regulatory DNA Design via Constrained RL

Xingyu Chen[1,2,3]*        Shihao Ma[1,2,3]*        Runsheng Lin[1]        Jiecong Lin[4]

Bo Wang[1,2,3]†

[1]University of Toronto
[2]Vector Institute for Artificial Intelligence
[3]University Health Network [4]Changping Laboratory

## Abstract

Designing regulatory DNA sequences that achieve precise cell-type-specific gene expression is crucial for advancements in synthetic biology, gene therapy and precision medicine. Although transformer-based language models (LMs) can effectively capture patterns in regulatory DNA, their generative approaches often struggle to produce novel sequences with reliable cell-type-specific activity. Here, we introduce *Ctrl-DNA*, a novel constrained reinforcement learning (RL) framework tailored for designing regulatory DNA sequences with controllable cell-type specificity. By formulating regulatory sequence design as a biologically informed constrained optimization problem, we apply RL to autoregressive genomic LMs, enabling the models to iteratively refine sequences that maximize regulatory activity in targeted cell types while constraining off-target effects. Our evaluation on human promoters and enhancers demonstrates that *Ctrl-DNA* consistently outperforms existing generative and RL-based approaches, generating high-fitness regulatory sequences and achieving state-of-the-art cell-type specificity. Moreover, *Ctrl-DNA*-generated sequences capture key cell-type-specific transcription factor binding sites (TFBS), short DNA motifs recognized by regulatory proteins that control gene expression, demonstrating the biological plausibility of the generated sequences.

Code available at: github.com/bowang-lab/Ctrl-DNA

## 1 Introduction

Cis-Regulatory elements (CRE), such as promoters and enhancers, are critical DNA sequences that control gene expression. The ability to engineer DNA sequences with precise regulatory activities has widespread implications in biotechnology, including gene therapy, synthetic biology, and precision medicine [1, 2]. A particularly desirable but challenging goal is designing CREs that drive high gene expression in a target cell type while maintaining controlled or limited fitness [3] in off-target cell types. A CRE's regulatory function is largely determined by its transcription factor binding sites (TFBSs), which are short DNA motifs recognized by transcription factors (TFs) that mediate gene regulation in cells. The presence or absence of specific TFBSs directly influences the fitness of a sequence across different cellular contexts. Although millions of regulatory sequences have evolved

---

*Equal contribution.

†Corresponding author: bo.wang@vectorinstitute.ai

[3]CRE fitness is defined as the ability to enhance gene expressions.

39th Conference on Neural Information Processing Systems (NeurIPS 2025).

naturally [3], these sequences are not optimized for targeted biomedical applications. For example, despite the human body comprising over 400 distinct cell types, very few cell-type-specific promoters have been identified [4]. This scarcity highlights the need for novel, engineered CREs with precise cell-type specificity. However, the design space for regulatory sequences is immense: a 100-base sequence yields approximately $4^{100}$ possibilities, making purely experimental approaches expensive and impractical.

Massively parallel reporter assays (MPRAs) have significantly advanced our ability to evaluate large libraries of DNA sequences for their cell-type-specific fitness [5, 6]. Building upon these assays, recent deep learning approaches leverage predictive models as reward functions to guide the optimization of CREs [1, 7]. However, these methods typically rely on iterative optimization strategies based on mutating existing or randomly initialized sequences, which often limits the sequence diversity and can trap optimization in local optima. Furthermore, enforcing complex constraints in multiple cell types is non-trivial in these frameworks.

Recent studies have adapted autoregressive language models (LMs) for regulatory DNA sequence design, successfully capturing functional sequence patterns and enabling the generation of sequences with desired properties, such as enhanced gene expression levels [8]. However, these models primarily imitate the distribution of known sequences, limiting their ability to explore novel regions of the sequence landscape. Reinforcement learning (RL) has emerged as an approach to finetune genomics LMs for optimizing DNA sequence design. However, existing RL-based approaches for cell-type-specific CRE design typically focus on maximizing fitness in a target cell type without accounting for fitness in other cell types [9]. To date, integrating explicit constraints within RL frameworks to suppress off-target regulatory activity using genomic LMs remains unexplored. Furthermore, conventional RL optimization strategies often depend on accurate value models and dense reward signals, introducing increased difficulty and inefficiency when navigating the vast DNA sequence space with complex biological constraints and sparse fitness reward signals.

To address these limitations, we introduce **Ctrl-DNA**, a reinforcement learning framework for controllable cell-type-specific regulatory DNA design via constrained RL. To the best of our knowledge, this work represents one of the first efforts to design regulatory DNA sequences with precise and controllable cell-type specificity. Building on recent advances in RL [10, 11, 12], we develop an RL-based fine-tuning framework based on pre-trained autoregressive genomic LMs. Our method avoids value model training by incorporating Lagrangian-regularized policy gradients directly from batch-normalized rewards, enabling stable and efficient optimization across multiple cell types. *Ctrl-DNA* supports explicit cell-type-specific constraints, enabling the generation of sequences with high expression in target cell types while constraining off-target activities. We evaluate *Ctrl-DNA* on human promoter and enhancer design tasks across six cell types. Our results show that *Ctrl-DNA* consistently outperforms existing generative and RL-based methods, achieving higher activity in target cell types while improving constraint satisfaction in non-target cell types. We also show that *Ctrl-DNA*-generated sequences maintain substantial sequence diversity and effectively capture biologically meaningful, cell-type-specific regulatory motifs.

Our main contributions are as follows:

- We develop a novel constraint-aware RL framework for CRE design, utilizing Lagrange multipliers explicitly and effectively to control cell-type specificity. To our knowledge, this represents one of the first efforts to incorporate constraint-based optimization into regulatory sequence generation.

- By directly computing policy gradients from batch-normalized biological rewards and constraints, our method eliminates the need for computationally expensive value models, enabling efficient learning under complex CRE design constraints.

- Through extensive empirical evaluations on human promoter and enhancer design tasks, we demonstrate that *Ctrl-DNA* consistently outperforms existing generative and RL-based methods, achieving higher targeted regulatory activity with state-of-the-art cell-type specificity.

## 2  Related Works

**DNA Sequence Design Optimization**: Optimization strategies complement generative models by explicitly steering sequences toward desired functions. Classical evolutionary algorithms, such as

genetic algorithms, iteratively refine sequences using fitness predictors, but they are often computationally expensive and may converge to suboptimal solutions [13]. To improve efficiency, heuristic techniques such as greedy search have been developed, incrementally editing sequences toward higher predicted performance [1]. Gradient-based approaches leverage differentiable surrogate models (e.g., neural predictors like Enformer) to perform gradient ascent directly in sequence space [4, 14]. Although computationally efficient, these methods often initialize from random or high-fitness observed sequences, reducing the diversity of generated sequences. Reinforcement learning (RL) offers a powerful framework that combines generative modeling with goal-directed optimization. DyNA-PPO [15] demonstrated the effectiveness of deep RL for DNA design, outperforming random mutation-based methods. GFlowNets further advanced this direction by learning stochastic policies that align with reward distributions, enabling diverse exploration of sequence space [16]. More recently, TACO [9] used RL to fine-tune pretrained DNA language models with biologically informed rewards. However, these approaches primarily focus on optimizing fitness in a single target cell type, without mechanisms to suppress or constrain activity in undesired cell types.

**Constrained Reinforcement Learning**: Constrained Reinforcement Learning (CRL), or often formulated as constrained Markov decision processes (CMDPs), addresses the critical challenge of optimizing policies under explicit constraints. Early foundational studies include actor-critic methods [17] and constrained policy optimization with function approximation [18]. Subsequent studies have explored integrating constraints into RL, often using Lagrangian methods that introduce non-stationarity into rewards [19, 20, 21]. Regularized policy optimization augments standard objectives with Kullback–Leibler (KL) or trust-region constraints [22], and is widely used in both single-task [23] and multi-task settings [24]. CRL has been applied in language generation to suppress undesired outputs through techniques like Lagrangian reward shaping [11, 12, 25], balancing primary objectives with safety constraints. However, such approaches remain largely unexplored in regulatory DNA design, where sparse rewards and multiple cell-type-specific constraints introduce significant challenges for standard constrained RL frameworks.

**Generative Models for Biological Sequence Design**: Deep generative models have advanced the design of functional DNA sequences. Diffusion-based approaches have emerged as promising tools, beginning with the Dirichlet Diffusion Score Model (DDSM), which creates promoters based on expression levels [26]. Building on this foundation, several subsequent studies have further developed diffusion models for designing regulatory DNA sequences[27, 28, 29]. Researchers have also leveraged Generative Adversarial Networks (GANs) for regulatory sequence design, with [7] creating cell-type-specific enhancers in Drosophila and humans, and ExpressionGAN [30] generates yeast promoters that outperform natural sequences in expression efficiency. Autoregressive genomic language models have recently been applied to model DNA sequences, learning statistical patterns from large-scale genomic datasets. For example, RegLM[8] fine-tuned the HyenaDNA model [31] using prefix tokens that encode expression levels, allowing the generation of enhancers with controlled activity. However, despite producing biologically plausible sequences, generative models typically replicate distributions observed in training data, constraining their ability to explore novel, out-of-distribution solutions. This inherent limitation underscores the need for integrating generative approaches with optimization frameworks such as reinforcement learning.

# 3 Methods

## 3.1 Problem Formulation

We formulate DNA sequence design as a constrained Markov decision process (CMDP). A DNA sequence is defined as $X = (x_1, x_2, \ldots, x_L) \in V^L$, where $V = \{A, C, G, T\}$ is the nucleotide vocabulary and $L$ is the sequence length. The CMDP is defined as $\mathcal{M} = (\mathcal{S}, \mathcal{A}, p, R_0, \{R_i\}_{i=1}^m, \{\delta_i\}_{i=1}^m)$, where $\mathcal{S}$ is the state space, $\mathcal{A} = V$ is the action space. $p(s_{t+1} \mid s_t, a_t)$ is a transition function that appends nucleotide $a_t$ to the current sequence prefix $s_t$. The sequence is evaluated by $m$ black-box reward functions $\{R_i : V^L \to \mathbb{R}\}_{i=0}^m$, where we denote reward $R_0$ as the CRE fitness in target cell, and reward $R_i$ for $i \geq 1$ as CRE fitness in off-target cell types. The values $\delta_i \in \mathbb{R}$ is the constraint threshold for off-target cell $i$. At each time step $t$, the agent observes state $s_t = (x_1, \ldots, x_{t-1}) \in \mathcal{S}$, selects an action $a_t \in \mathcal{A}$ according to a policy $\pi_\theta(a_t \mid s_t)$, and transitions to the next state $s_{t+1}$. Rewards $\{R_i(X)\}$ are only calculated at the terminal step $t = L$.

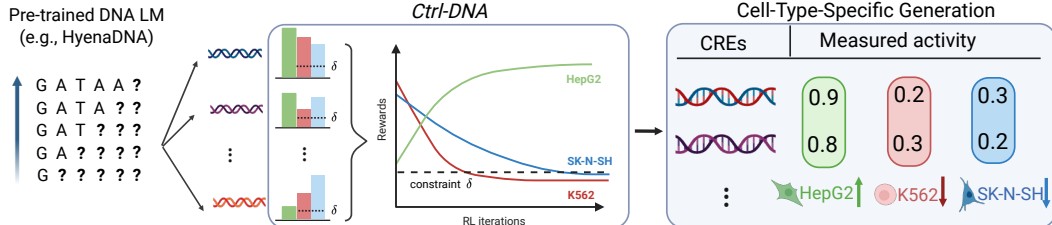

Figure 1: Overview of the *Ctrl-DNA* framework for controllable regulatory sequence generation. *Ctrl-DNA* builds on a pre-trained autoregressive DNA language model and applies constrained reinforcement learning to guide sequence generation toward high fitness in a target cell type (e.g., HepG2) while suppressing off-target fitness (e.g., K562, SK-N-SH), enabling the generation of CREs with strong cell-type specificity.

Our objective is to learn a policy $\pi_\theta$ that maximizes the expected CRE fitness in the target cell type while ensuring off-target fitness remains within the specified constraints. Formally, we aim to solve:

$$\max_{\pi_\theta} \; \mathbb{E}_{X \sim \pi_\theta}[R_0(X)] \quad \text{s.t.} \quad \mathbb{E}_{X \sim \pi_\theta}[R_i(X)] \leq \delta_i, \quad \forall i \in \{1, \dots, m\}. \tag{1}$$

For clarity, we define $J_i(\theta) = \mathbb{E}_{X \sim \pi_\theta}[R_i(X)]$ as the expected reward for cell type $i$, where $J_0(\theta)$ is referred to as task rewards and $J_i(\theta)$ for $i \geq 1$ is referred to as constraints (also called off-target rewards throughout this paper).

### 3.2 Constrained Batch-wise Relative Policy Optimization

We now describe our approach for solving the constrained reinforcement learning problem for CRE sequence generation introduced in Section 3.1. Most deep-learning-based constrained RL methods rely on training one or more value models to estimate expected returns and costs [11, 32, 12], which can significantly increase training complexity. Moreover, reward signals that are sparse and only available at the end of a generated sequence may further complicate the training of value models that need accurate values at each step [33].

To address this, we adapt work in [33, 10] for our constrained DNA sequence design task, avoiding value network training for each cell type while enforcing constraints on off-target cell CRE fitness. We adopt a primal-dual approach based on Lagrangian relaxation, which introduces adaptive multipliers to enforce constraints while optimizing the main objective.

**Lagrangian Relaxation and Constrained RL.** The Lagrangian relaxation of the constrained objective in Eq. 1 defines a primal-dual optimization problem:

$$\mathcal{L}_{\text{lag}}(\theta, \lambda) = \max_\theta \min_{\lambda \geq 0} \left[ J_0(\theta) - \sum_{i=1}^m \lambda_i (J_i(\theta) - \delta_i) \right], \tag{2}$$

where $\delta_i$ is a user-specified threshold and $\lambda_i \geq 0$ is a dual variable for constraint $i$.

In practice, we solve primal-dual policy optimization by taking iterative gradient ascent-descent steps of the policy parameter $\theta$ and Lagrange multiplier $\lambda_i$:

$$\theta_{k+1} = \theta_k + \eta_\theta \nabla_\theta \mathcal{L}_{\text{lag}}(\theta, \lambda) = \theta_k + \eta_\theta \nabla_\theta \left[ J_0(\theta) - \sum_{i=1}^m \lambda_i J_i(\theta) \right], \tag{3}$$

$$\lambda_{i,k+1} = \lambda_{i,k} - \eta_{\lambda_i} \nabla_{\lambda_i} \mathcal{L}_{\text{lag}}(\theta, \lambda) = \lambda_k - \eta_{\lambda_i} \nabla_{\lambda_i} \left[ \lambda_i (\delta_i - J_i(\theta)) \right]. \tag{4}$$

where $\eta_\theta$ and $\eta_{\lambda_i}$ are learning rates. $k$ denotes the optimization step. This min-max formulation seeks a saddle point that maximizes reward while satisfying constraints [21, 11].

**Our Methods.** As commonly done in reinforcement learning [34, 35], $\nabla_\theta J(\theta)$ is calculated by policy gradient methods where $\nabla_\theta J(\theta) = \mathbb{E}_\pi [\Psi_t \nabla_\theta \log \pi_\theta(a_t \mid s_t)]$. $\Psi_t$ represents a surrogate signal such as rewards, state-action values or advantage estimates [36]. While standard approaches compute advantages using learned value functions, we avoid value network training by drawing

inspirations from the Group Relative Policy Optimization (GRPO) framework [10, 33]. GRPO estimates advantages by comparing outputs generated from the same prompt. In contrast, we propose a batch-level variant for CRE sequence optimization, where advantages are computed by grouping sequences within each training batch.

Formally, for each objective $i \in \{0, \dots, m\}$, the normalized advantage for sequence $X_j$ is defined as $A_i^{(j)} = \frac{R_i(X_j) - \bar{R}_i}{\sigma(R_i)}$, where $R_i(X_j)$ is the reward assigned by the $i$-th reward function, and $\bar{R}_i$, $\sigma(R_i)$ denote the batch mean and standard deviation of $R_i$ over the current batch of sequences. To guide policy updates under constraints, we use the Lagrange multipliers to form a convex combination of advantages from different cell types. We define the Lagrangian advantage as:

$$\hat{A}^{(j)} = \left( m - \sum_{i=1}^{m} \lambda_i \right) A_0^{(j)} - \sum_{i=1}^{m} \lambda_i A_i^{(j)}, \tag{5}$$

where $m$ is the number of constraints, and $\lambda_i^{(j)}$ is the Lagrange multiplier applied for constraint $i$. This encourages the policy to favor sequences with high target rewards while discouraging those that violate constraints.

To estimate $\nabla_\theta J(\theta)$ during policy updates, we adopt a clipped surrogate objective with KL regularization [10, 35]:

$$\mathcal{L}_{\text{policy}}(\theta) = \frac{1}{B} \sum_{j=1}^{B} \sum_{i=1}^{T} \min \left\{ \rho_i^{(j)} \hat{A}^{(j)}, \ \text{clip}_\epsilon(\rho_i^{(j)}) \hat{A}^{(j)} \right\} - \beta \cdot \text{KL}(\pi_\theta \,||\, \pi_{\text{ref}}), \tag{6}$$

where $\pi_\theta$ and $\pi_{\text{old}}$ denote the current and previous policy networks. $\pi_{\text{ref}}$ is the reference model, which usually is the initial policy model. Here, $\rho_i^{(j)} = \frac{\pi_\theta(a_i^j | s_i^j)}{\pi_{\text{old}}(a_i^j | s_i^j)}$ is the importance sampling ratio, and $\text{clip}_\epsilon(\rho_i^{(j)}) = \text{clip}(\rho_i^{(j)}, 1 - \epsilon, 1 + \epsilon)$ restricts large policy updates. The coefficient $\beta$ controls the strength of the KL divergence penalty, and $\epsilon$ sets the clipping threshold.

To adaptively enforce constraints, we update the Lagrange multiplier $\lambda_i$ based on batch-level constraint satisfaction. For each constraint $i$, we define the multiplier loss as $\mathcal{L}_{\text{multiplier}}(\lambda_i) = \frac{1}{B} \sum_{j=1}^{B} \left( R_i^{(j)} - \delta_i \right) \lambda_i$, where $\delta_i$ is the constraint for cell type $i$. This formulation increases the penalty on off-target cell types whose predicted fitness exceeds the constraint thresholds, while reducing the weight of those that already satisfy the constraints.

With this setup, the primal-dual updates (Equation 3 & 4) become:

$$\theta_{k+1} = \theta_k + \eta_\theta \nabla_\theta \mathcal{L}_{\text{policy}}(\theta), \quad \lambda_{i,k+1} = \lambda_{i,k} - \eta_{\lambda_i} \nabla_{\lambda_i} \mathcal{L}_{\text{multiplier}}(\lambda_i).$$

Detailed pseudocode for the full algorithm and gradient functions are provided in Appendix A.1.

**Empirical Designs.** To improve training stability and model performance, we introduce several empirical modifications. First, we maintain a replay buffer of previously generated sequences and mix them with samples from the current policy. This helps reduce variance in batch-level reward statistics and leads to smoother advantage estimation. Second, we clip each Lagrange multiplier $\lambda_i$ to the range $[0, 1]$, which prevents overly aggressive constraint enforcement and stabilizes the dual updates. Lastly, to prevent the main objective from being overwhelmed when constraint weights are large, we clip its coefficient in Eq. 5 as $\min(1, m - \sum_{i=1}^{m} \lambda_i)$, ensuring sufficient signal for optimizing the target reward.

### 3.3 Regularizing Generated Sequences via TFBS Frequency Correlation

Although *Ctrl-DNA* effectively optimizes sequence generation under specified constraints, the resulting sequences may still deviate from biologically realistic distributions due to reward hacking [37]. To further regularize the generated distribution toward biologically plausible patterns, we introduce an additional reward term based on the correlation between transcription factor binding site (TFBS) frequencies in generated and real sequences.

Specifically, we first compute TFBS frequencies from a reference set of real DNA sequences. For each TFBS, we calculate its occurrence frequency across these real sequences, forming a reference

frequency vector $q_{\text{real}}$. Similarly, for each sequence generated by the policy $\pi_\theta$, we compute a corresponding motif frequency vector $q_{\text{gen}}$. We then quantify the similarity between generated and real sequence distributions using the Pearson correlation coefficient: $R_{\text{TFBS}}(X) = \text{Corr}(q_{\text{gen}}, q_{\text{real}})$ for each generated sequence $X$.

We treat $R_{\text{TFBS}}$ as an additional constraint reward function to maintain realistic TFBS patterns in generated sequences. However, to prevent the policy from overfitting to correlation alone and generating sequences that merely replicate the real distribution, we apply a clipped upper bound on the corresponding dual multiplier $\lambda_{\text{TFBS}}$. That is, from Equation 4, we clip $\lambda_{TFBS}$ to the range $[0, \lambda_{max}]$ where $\lambda_{max} \leq 1$, where $\lambda_{\max}$ is a predefined hyperparameter. This clipping mechanism ensures a balanced optimization process that maintains realistic TFBS frequencies without overly constraining policy exploration or the main optimization objective.

TFBS information is widely used in CRE design [8, 4, 29]. However, existing methods typically incorporate TF motifs either as post-hoc evaluation metrics or through explicit tiling strategies during sequence design. A closely related approach is TACO [9], which trains a LightGBM model to predict sequence fitness from motif frequencies and derives motif-level rewards from SHAP values. In contrast, our method bypasses the need for additional predictive models by directly aligning the motif frequency distribution of generated sequences with that of real sequences. This removes the potential biases introduced by model training and provides a more reliable regularization signal from real biological distributions.

## 4 Experiments

### 4.1 Experimental Setup

**Datasets.** We evaluate our method on human promoter and enhancer datasets [38, 39]. The enhancer dataset includes 200 bp sequences from three cell lines: HepG2, K562, and SK-N-SH. The promoter dataset consists of 250 bp sequences from Jurkat, K562, and THP1. Each dataset contains sequence-fitness pairs across the three respective cell types, with fitness measured via massively parallel reporter assays (MPRAs) [40]. We adopt the preprocessing pipeline described in [8] to preprocess all datasets. Please refer to Appendix A.4 for details.

**Motif Processing and Real Sequence Statistics.** We obtain human-specific position probability matrices (PPMs) and pairwise motif correlation data from the JASPAR 2024 database [41]. Following the preprocessing procedure described in [8], we retain a curated set of 464 human transcription factor motifs. For each task, we identify real sequences by selecting those in the top 50th percentile for fitness in the target cell type and the bottom 50th percentile in off-target cell types. We then apply FIMO [42] to scan for motif occurrences and compute motif frequency distributions, which serve as the reference motif distribution for optimization and evaluation.

**Reward Models.** We train a reward model for each cell type using the Enformer architecture [43], following protocols from prior works on cis-regulatory element (CRE) design [8, 9, 4, 29, 38]. For the human enhancer task, we directly adopted the pretrained model weights from [8]. For human promoters, we trained a separate reward model using the data split from [4] (70% train, 20% validation, 10% test). Training was conducted using the AdamW optimizer with a learning rate of 1e-4 and MSE loss, over 20 epochs. The checkpoint with the best validation performance was used for evaluation.

All reward models are based on an Enformer architecture [44], combining convolutional and Transformer layers, which has shown strong performance in DNA regulatory prediction tasks. Both the enhancer and promoter models share the same architecture: 11 layers with a hidden dimension of 1536.

**Models and Baselines.** For sequence generation, we fine-tune HyenaDNA [31], an autoregressive genomic LM pre-trained on the human genome, as our policy model. We compare our proposed method, *Ctrl-DNA*, against a diverse set of baselines, including evolutionary algorithms (AdaLead [45], BO [46], CMAES [47], PEX [48]), generative models (RegLM [8]), and reinforcement learning approaches (TACO [9], PPO [35], PPO-Lagrangian [12]). Full baseline details are provided in Appendix A.5.

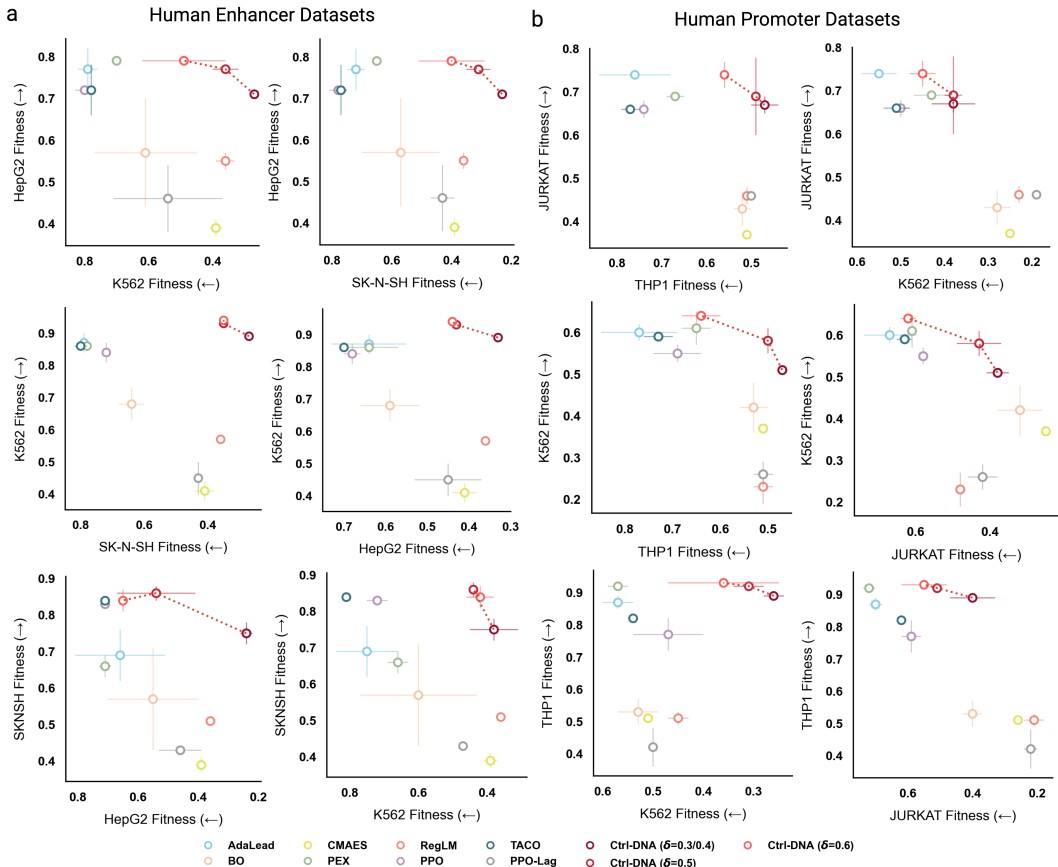

Figure 2: Pairwise fitness comparison of generated CREs highlights *Ctrl-DNA*'s cell-type specificity. Each subplot compares mean ± s.d. fitness in two human cell lines (y = target, x = off-target); points in the **top-right** denote sequences with high on-target and low off-target fitness. Baseline methods are shown in pastel colors, while *Ctrl-DNA* variants ($\delta$ = 0.3/0.4, 0.5, 0.6) are connected in red dotted lines, illustrating the trade-off as constraint strength increases and *Ctrl-DNA*'s dominance in the top-right corner for both enhancer (a) and promoter (b) datasets.

**Evaluation Metrics.** To assess the performance of each method, we report median rewards ( **Median**) over the generated sequences in the final round. Additionally, we report **Reward Difference** ($\Delta R$) to quantify the average difference between the target cell's fitness and the fitness across off-target cell types, indicating the cell-type fitness specificity. **Motif Correlation** is defined as the Pearson correlation between TFBS frequencies in generated sequences and real sequences. A higher correlation indicates greater alignment with biologically plausible motif distributions. Lastly, **Diversity** is calculated as the Shannon entropy of the generated sequences in the final round, reflecting the model's ability to explore diverse solutions. We report the mean and standard deviation of each metric over 5 runs initialized with different random seeds.

## 4.2 *Ctrl-DNA* optimizes enhancer and promoter sequences under cell-type-specific constraints

We evaluate *Ctrl-DNA* on two regulatory sequence design tasks using the Human Enhancer and Human Promoter datasets. Results are presented in Figure 2. In these plots, points positioned further to the right indicate lower fitness in off-target cell types (i.e., better constraint satisfaction), while points higher on the vertical axis indicate higher fitness in the target cell type. Methods that appear in the upper-right corner achieve the best trade-off between maximizing target cell fitness and minimizing off-target expression.

Table 1: Performance comparison across methods for each target cell type on the Human Enhancer and Human Promoter datasets. For each target, we report $\Delta R$ ($\uparrow$) and motif correlation ($\uparrow$). Constraint thresholds are set to 0.5 for all six cell types. Note that K562[*] refers to the K562 cell type in the Human Promoter dataset. Motif Corr[†] computed using 90th-percentile reference sequences.

| Cell Type | Metric | AdaLead | BO | CMAES | PEX | RegLM | PPO | TACO | PPO-Lag | Ctrl-DNA |
|---|---|---|---|---|---|---|---|---|---|---|
| HepG2 | $\Delta R \uparrow$ | 0.02 (0.04) | -0.03 (0.03) | 0.01 (0.01) | 0.13 (0.03) | 0.16 (0.01) | -0.06 (0.02) | -0.05 (0.01) | -0.02 (0.03) | **0.49 (0.01)** |
|  | Motif Corr $\uparrow$ | **0.45 (0.03)** | 0.21 (0.10) | 0.15 (0.08) | 0.31 (0.04) | 0.22 (0.07) | 0.41 (0.14) | 0.30 (0.03) | 0.41 (0.02) | **0.43 (0.07)** |
| K562 | $\Delta R \uparrow$ | 0.16 (0.05) | 0.08 (0.02) | 0.03 (0.02) | 0.17 (0.03) | 0.19 (0.02) | 0.14 (0.04) | 0.11 (0.01) | 0.05 (0.05) | **0.54 (0.01)** |
|  | Motif Corr $\uparrow$ | 0.49 (0.07) | 0.08 (0.18) | 0.06 (0.04) | 0.21 (0.06) | 0.23 (0.01) | 0.35 (0.11) | 0.28 (0.02) | 0.44 (0.05) | **0.51 (0.02)** |
| SK-N-SH | $\Delta R \uparrow$ | -0.02 (0.07) | -0.01 (0.02) | 0.00 (0.01) | -0.04 (0.01) | 0.14 (0.01) | -0.04 (0.08) | 0.08 (0.02) | -0.04 (0.07) | **0.37 (0.11)** |
|  | Motif Corr $\uparrow$ | 0.15 (0.13) | 0.05 (0.06) | 0.03 (0.04) | 0.17 (0.02) | 0.18 (0.01) | 0.23 (0.01) | 0.11 (0.12) | 0.42 (0.03) | **0.25 (0.04)** |
| JURKAT | $\Delta R \uparrow$ | 0.09 (0.06) | 0.04 (0.03) | -0.00 (0.01) | 0.15 (0.03) | 0.09 (0.01) | 0.04 (0.02) | 0.03 (0.12) | 0.11 (0.01) | **0.25 (0.01)** |
|  | Motif Corr $\uparrow$ | 0.41 (0.07) | 0.19 (0.23) | 0.30 (0.02) | 0.60 (0.05) | 0.14 (0.02) | 0.61 (0.11) | 0.55 (0.11) | 0.29 (0.33) | **0.69 (0.02)** |
| K562[*] | $\Delta R \uparrow$ | -0.12 (0.01) | -0.15 (0.02) | -0.17 (0.02) | -0.04 (0.02) | -0.08 (0.03) | -0.09 (0.03) | -0.10 (0.02) | -0.22 (0.04) | **0.12 (0.02)** |
|  | Motif Corr $\uparrow$ | 0.60 (0.15) | 0.13 (0.16) | 0.24 (0.08) | 0.63 (0.02) | 0.42 (0.04) | 0.39 (0.11) | 0.50 (0.12) | 0.42 (0.22) | **0.75 (0.06)** |
| THP1 | $\Delta R \uparrow$ | 0.24 (0.01) | 0.20 (0.02) | 0.20 (0.01) | 0.29 (0.01) | -0.01 (0.01) | 0.24 (0.03) | 0.24 (0.08) | 0.18 (0.03) | **0.56 (0.01)** |
|  | Motif Corr $\uparrow$ | 0.63 (0.06) | 0.26 (0.09) | 0.19 (0.06) | **0.84 (0.01)** | 0.35 (0.01) | 0.42 (0.10) | 0.36 (0.03) | 0.42 (0.10) | 0.25 (0.04) |
|  | Motif Corr[†] $\uparrow$ | 0.16 (0.13) | 0.06 (0.08) | 0.06 (0.04) | 0.04 (0.02) | 0.29 (0.01) | 0.37 (0.04) | 0.33 (0.01) | -0.02 (0.07) | **0.60 (0.02)** |

For enhancers, we evaluate performance under three constraint thresholds ($\delta = 0.3, 0.5, 0.6$). Across all thresholds, *Ctrl-DNA* consistently achieves the highest target-cell fitness while satisfying the off-target constraints. PPO-Lagrangian (PPO-Lag) struggles to balance optimization and constraint satisfaction, likely due to the difficulty of training value networks under sparse, sequence-level reward signals. Notably, while methods such as TACO and CMAES achieve relatively high expression in the target cell type, they fail to suppress off-target fitness, leading to poor cell-type specificity.

The promoter design task is a more challenging task because all three target cell types are mesoderm-derived hematopoietic cells, which share substantial transcriptional similarity [4]. We test under three constraint thresholds ($\delta = 0.4, 0.5, 0.6$). *Ctrl-DNA* outperforms all baselines in maximizing target cell-type fitness and satisfying constraints at $\delta = 0.5$ and 0.6. However, no method, including *Ctrl-DNA*, successfully reduces THP1 fitness below the stricter threshold of $\delta = 0.4$. We hypothesize that this is due to the data distribution: the 25th percentile of THP1 fitness is already 0.49 (Appendix A.4), indicating that most sequences exhibit high expression in this cell type. Despite this challenge, when THP1 is treated as an off-target cell, *Ctrl-DNA* still achieves the lowest THP1 fitness among all methods.

Across both enhancer and promoter tasks, *Ctrl-DNA* consistently achieves the best trade-off between optimizing target cell types and enforcing cell-type-specific constraints, substantially outperforming existing RL and generative baselines. Interestingly, we observe a clear trade-off when enforcing stricter constraints: as the constraint threshold decreases, the fitness in the target cell type slightly declines. This trend likely arises because stricter constraint enforcement potentially narrows the feasible sequence space, making it more challenging to simultaneously optimize target cell-type activity and minimize off-target expression. For completeness, we compared our constrained formulation to a scalarized differential-expression (DE) objective and found that DE imposes a fixed trade-off that underperforms our adaptive Lagrangian approach; see Appendix A.8 for details. Despite this inherent difficulty, *Ctrl-DNA*'s constraint-aware optimization framework remains highly effective, demonstrating robustness in maintaining superior target-cell fitness even under rigorous constraint conditions.

### 4.3 *Ctrl-DNA* captures biologically relevant motifs with higher specificity

Besides the fitness of generated CRE sequences in each cell type, we also evaluate the sequences with three other metrics: reward difference ($\Delta R$), motif correlation and diversity. In Table 1, we can observe that *Ctrl-DNA* achieves highest reward differences across all cell types in both human promoter and enhancers, indicating it is better at optimizing DNA sequences for cell-type-specific fitness. For motif correlation, *Ctrl-DNA* also achieves stronger performance across most cell types, except for THP1 in promoter design. As noted in Section 4.2, THP1 fitness values are skewed, with the majority of sequences in the dataset exhibiting fitness around 0.5. Since motif correlation is evaluated against sequences near the 50th percentile (see Section 4.1), the resulting motif frequency distribution may not accurately reflect the high-activity sequences we aim to design.

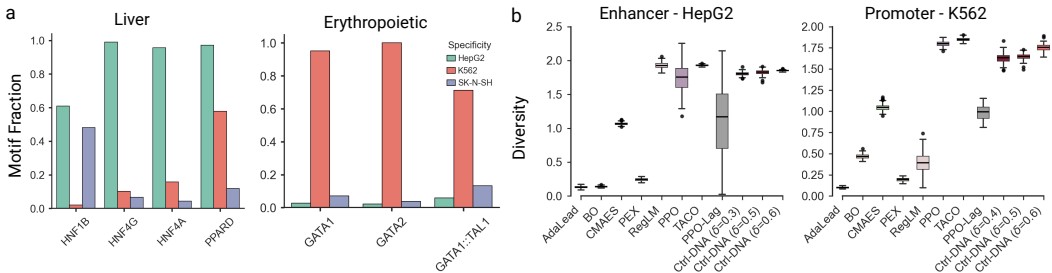

Figure 3: (a) Fraction of Ctrl-DNA-generated enhancers containing selected cell-type-specific transcription factor (TF) motifs. (b) Diversity scores of generated sequences for HepG2 enhancers (left) and K562 promoters (right) across different methods.

To further investigate this discrepancy, we extract motifs from promoter sequences in the 90th percentile of THP1 fitness, applying a significance threshold of $q < 0.05$ to avoid false positives. We then re-evaluate the motif correlation between generated sequences and this more selective reference set. These results, denoted as Motif Corr[†] in Table 1, show that *Ctrl-DNA* outperforms all baselines under this stricter setting. In contrast, most baseline methods exhibit reduced motif correlation, suggesting that they tend to align with non-informative or broadly distributed motifs. Despite being regularized using motifs from a less selective reference set, *Ctrl-DNA* successfully prioritizes the most discriminative motifs during optimization. Finally, the correlation improvements persist when evaluated against alternative reference sets explicitly designed to reduce circularity (Appendix A.10), with consistently high correlations across cell lines.

To further demonstrate that *Ctrl-DNA* selects more cell-type-discriminative motifs during sequence generation, we evaluated the frequency of known cell-type-specific TFBS in the generated sequences. In particular, we examined generated sequences for liver-specific and erythropoietic-specific motifs. As shown in Figure 3a, *Ctrl-DNA*-generated sequences for HepG2 (a liver-derived cell line) show the highest frequency of liver-specific motifs such as HNF4A and HNF4G. Similarly, sequences generated for K562 (an erythropoietic lineage cell line) contain the highest frequency of erythropoietic-specific motifs such as GATA1 and GATA2. These findings indicate that *Ctrl-DNA* not only optimizes for target-cell fitness, but also learns regulatory patterns that reflect underlying cell-type specificity.

Finally, we assess the diversity of generated sequences in Figure 3b and Appendix A.6. *Ctrl-DNA* achieves comparable or higher diversity than most baselines, indicating its ability to generate diverse sequences without sacrificing regulatory control.

### 4.4 Ablation Study

**Constraint Formulations.** To investigate alternative constraint enforcement strategies, we explored several other constrained methods from current works. First, we adapt the loss from Interior-point Policy Optimization (IPO) [49], referring to this variant as *Ctrl-DNA-IPO*. Second, we implement a log-barrier penalty on constraint rewards following [50], which we denote as *Ctrl-DNA-Log*. All experiments are conducted using a constraint threshold of $0.5$. See Appendix A.7 for detailed setup. As shown in Table 2, *Ctrl-DNA-Log* suppresses off-target rewards effectively but fails to maintain high fitness in the target cell type. In contrast, *Ctrl-DNA-IPO* improves target reward but does not enforce constraints adequately. These results highlight that our proposed formulation strikes a better balance between optimizing target cell-type fitness and minimizing off-target expression.

**TFBS Regularization.** In Section 3.3, we introduced a correlation-based regularization using TFBS motif frequencies to promote biologically plausible sequences. By changing the upper bound on the TFBS multiplier ($\lambda_{\max}$) we can limit the weight we put on this regularization. From Table 2, we observe that increasing $\lambda_{\max}$ from 0.0 to 0.1 improves motif correlation without substantially degrading other metrics. In certain cell types, such as JURKAT, a higher value of $\lambda_{\max}$ also leads to improved optimization performance (See Appendix Table 7). This supports the utility of TFBS regularization in guiding sequence generation. However, since our comparisons use motif frequencies computed from a loosely matched reference set, we recommend tuning $\lambda_{\max}$ carefully in practical applications depending on the reliability of the available ground truth.

Table 2: Ablation study on constraint formulation and TFBS regularization. We compare variants of *Ctrl-DNA* using alternative constraint handling methods (*Ctrl-DNA-IPO*, *Ctrl-DNA-Log*) and varying TFBS regularization strengths ($\lambda_{\max}$). Results are reported on the Human Enhancer dataset (target cell: HepG2), with constraint threshold $\delta = 0.5$ in both experiments. See Appendix A.7 for complete results across all datasets.

| Method | Target: HepG2 | | | | | |
|---|---|---|---|---|---|---|
| | HepG2 ↑ | K562 ↓ | SK-N-SH ↓ | $\Delta R$ | Motif Correlation | Diversity |
| Ctrl-DNA-Log | 0.24 (0.02) | 0.24 (0.03) | 0.21 (0.04) | 0.02 (0.06) | 0.16 (0.13) | 1.62 (0.08) |
| Ctrl-DNA-IPO | 0.74 (0.01) | 0.86 (0.02) | 0.83 (0.02) | -0.10 (0.02) | 0.39 (0.14) | 1.58 (0.13) |
| Ctrl-DNA ($\lambda_{max} = 0.00$) | 0.78 (0.02) | 0.40 (0.06) | 0.33 (0.03) | 0.42 (0.02) | 0.33 (0.12) | 1.82 (0.02) |
| Ctrl-DNA ($\lambda_{max} = 0.01$) | 0.77 (0.01) | 0.34 (0.04) | 0.30 (0.02) | 0.45 (0.02) | 0.16 (0.10) | 1.84 (0.01) |
| Ctrl-DNA ($\lambda_{max} = 0.1$) | 0.77 (0.01) | 0.36 (0.04) | 0.31 (0.04) | 0.44 (0.03) | 0.43 (0.07) | 1.82 (0.04) |

## 5 Discussion

Designing cell-type-specific cis-regulatory sequences presents a challenging optimization problem that involves balancing competing objectives. Our proposed method, *Ctrl-DNA*, achieves strong performance across both enhancer and promoter datasets, outperforming evolutionary and RL baselines in maximizing target cell-type fitness while satisfying off-target constraints. In addition, *Ctrl-DNA* supports explicit control over constraint thresholds, enabling flexible and controllable CRE sequence design. By computing Lagrangian advantages directly from batch-normalized rewards without training value models, *Ctrl-DNA* offers a lightweight and effective solution to constrained CRE DNA sequence generation.

Nevertheless, the ability to enforce constraints is inherently limited by the data distribution. For instance, in the THP1 promoter dataset, a large proportion of sequences exhibit high baseline activity, making it difficult to enforce stricter constraints such as $\delta = 0.4$. This challenge affects both the accuracy of the learned reward model and the capacity of *Ctrl-DNA* to suppress expression in such settings. These observations highlight the importance of considering dataset-specific characteristics when setting constraint thresholds or evaluating constrained RL methods.

Although *Ctrl-DNA* already demonstrates robust performance, there are several directions for improvement. First, tuning Lagrange multipliers remains empirical. Future work could explore adaptive control methods such as proportional–integral–derivative controllers [51]. Second, additional biological constraints could be incorporated to further improve the plausibility and functionality of generated sequences. Finally, our current framework is limited to reinforcement learning fine-tuning on autoregressive models. As a next step, we plan to extend *Ctrl-DNA* to other structures such as diffusion-based DNA models.

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

# A    Technical Appendices and Supplementary Material

## A.1    Training Details for *Ctrl-DNA*

We provide a high-level overview of our constrained reinforcement learning procedure for optimizing constraint-aware regulatory sequence generation. We also present the hyperparameter settings used in our experiments in Table 3. All models were trained using the Adam optimizer with a learning rate specified in Table 3 with 100 training epochs.

Experiments were conducted on a single NVIDIA A100 GPU with 40 GB of memory. Each experiment typically required between 1 to 2 hours of wall-clock time. The experiment results reported are mean performances across five seeds.

---

**Algorithm 1** Ctrl-DNA

---

**Require:**  Initialized policy $\pi_\theta$, Reference policy $\pi_{\text{ref}}$, Lagrange multipliers $\{\lambda_i\}_{i=1}^m$, reward functions $\{R_i\}_{i=0}^m$, reference TFBS frequency $q_{\text{real}}$, constraint thresholds $\{\delta_i\}$, learning rates $\eta_\theta$, $\eta_\lambda$, hyperparameters $\beta$, $\epsilon$, replay buffer $\mathcal{B}$, batch size $B$, replay batch size $B_r$

1: **for** each training iteration **do**
2:       Update $\pi_{\text{old}} = \pi_\theta$
3:       Sample $B$ sequences $\{X_j\}_{j=1}^B$ from policy $\pi_{\text{old}}$
4:       Compute rewards $\{R_i(X_j)\}$ for $i = 0, \ldots, m$
5:       Compute TFBS frequency $q_{\text{gen}}^{(j)}$ and correlation $R_{\text{TFBS}}(X_j) = \text{Corr}(q_{\text{real}}, q_{\text{gen}}^{(j)})$.
6:       Treat $R_{\text{TFBS}}$ as additional reward $R_{m+1}$ and append to the reward set
7:       Sample $B_r$ sequences from replay buffer $\mathcal{B}$ and merge with current batch
8:       Compute normalized advantage: $A_i^{(j)} = \frac{R_i(X_j) - \bar{R}_i}{\sigma(R_i)}$ for each reward $R_i$
9:       Compute clipped main reward coefficient:
10:           $\alpha_0 = \min(1, m - \sum_{i=1}^m \lambda_i)$
11:       Construct mixed advantage:
12:           $\hat{A}^{(j)} = \alpha_0 \cdot A_0^{(j)} - \sum_{i=1}^m \lambda_i A_i^{(j)}$
13:       Add current batch $\{(X_j, \{r_i^{(j)}\})\}$ to replay buffer $\mathcal{B}$
14:       **for** each policy update step **do**
15:           Update policy parameters $\theta$ using $\mathcal{L}_{\text{policy}}(\theta)$ (Eq. 6)
16:           **for** each constraint $i = 1, \ldots, m + 1$ **do**
17:               Update $\lambda_i$ using:
18:                   $\mathcal{L}_{\text{multiplier}}(\lambda_i) = \frac{1}{B+B_r} \sum_{j=1}^{B+B_r} (R_i(X_j) - \delta_i) \cdot \lambda_i$
19:           **end for**
20:       **end for**
21: **end for**

---

Table 3: Experiment Hyperparameters.

| Hyperparameter | HepG2 | K562 | SK-N-SH | JURKAT | K562 | THP1 |
|---|---|---|---|---|---|---|
| Batch Size | 256 | 256 | 256 | 256 | 256 | 256 |
| Replay Buffer Batch Size | 24 | 24 | 24 | 24 | 24 | 24 |
| Policy Learning Rate ($\eta_\theta$) | 1e-4 | 1e-4 | 1e-4 | 1e-4 | 1e-4 | 1e-4 |
| Multiplier Learning Rate ($\eta_\lambda$) | 3e-4 | 3e-4 | 3e-4 | 3e-4 | 3e-3 | 3e-3 |
| KL Value Coefficient ($\beta$) | 0.2 | 0.2 | 0.2 | 0.2 | 0.2 | 0.2 |
| TFBS Multiplier Upper Bound ($\lambda_{max}$) | 0.1 | 0.1 | 0.1 | 0.1 | 0.1 | 0.1 |

## A.2    Policy Optimization Objective

As discussed in Section 3.2, the objective for updating policy parameters $\theta$ is defined as:

$$\mathcal{L}_{\text{policy}}(\theta) = \frac{1}{B} \sum_{j=1}^B \sum_{i=1}^T \min\left\{ \rho_i^{(j)} \hat{A}^{(j)}, \; \text{clip}_\epsilon(\rho_i^{(j)}) \hat{A}^{(j)} \right\} - \beta \cdot \text{KL}(\pi_\theta \,\|\, \pi_{\text{ref}}), \tag{7}$$

where $\rho_i^{(j)} = \frac{\pi_\theta(a_i^j | s_i^j)}{\pi_{\text{old}}(a_i^j | s_i^j)}$ is the importance sampling ratio, and $\text{clip}_\epsilon(\rho_i^{(j)}) = \text{clip}(\rho_i^{(j)}, 1 - \epsilon, 1 + \epsilon)$ applies clipping for stability. Following [33], we assume one policy update per iteration, allowing $\pi_{\text{old}} = \pi_\theta$ for simplification.

The gradient of this objective with respect to $\theta$ becomes:

$$
\begin{aligned}
\nabla_\theta \mathcal{L}_{\text{policy}}(\theta) &= \frac{1}{B} \sum_{j=1}^{B} \sum_{i=1}^{T} \hat{A}^{(j)} \nabla_\theta \log \pi_\theta(a_i^{(j)} | s_i^{(j)}) \\
&\quad - \beta \cdot \nabla_\theta \left( \frac{\pi_{\text{ref}}(a_i^{(j)} | s_i^{(j)})}{\pi_\theta(a_i^{(j)} | s_i^{(j)})} - \log \frac{\pi_{\text{ref}}(a_i^{(j)} | s_i^{(j)})}{\pi_\theta(a_i^{(j)} | s_i^{(j)})} - 1 \right) \\
&= \frac{1}{B} \sum_{j=1}^{B} \sum_{i=1}^{T} \left[ \hat{A}^{(j)} - \beta \left( \frac{\pi_{\text{ref}}(a_i^{(j)} | s_i^{(j)})}{\pi_\theta(a_i^{(j)} | s_i^{(j)})} - 1 \right) \right] \nabla_\theta \log \pi_\theta(a_i^{(j)} | s_i^{(j)}).
\end{aligned}
\tag{8}
$$

### A.3 Lagrange Multiplier Update

We apply Lagrangian relaxation to enforce soft constraints on off-target cell types. The gradient of the multiplier objective with respect to each Lagrange multiplier $\lambda_i$ is given by:

$$
\nabla_{\lambda_i} \mathcal{L}_{\text{multiplier}}(\lambda_i) = \frac{1}{B} \sum_{j=1}^{B} \left( R_i(X_j) - \delta_i \right),
$$

where $R_i(X_j)$ is the constraint-specific reward (e.g., off-target activity) for sample $X_j$, and $\delta_i$ is the user-defined constraint threshold.

### A.4 Dataset

In this section, we describe the datasets used in our experiments. The human enhancer dataset contains cis-regulatory element (CRE) activity measured by MPRA across three cell lines: HepG2 (liver cell line), K562 (erythrocyte cell line), and SK-N-SH (neuroblastoma cell line). Each sequence in this dataset is 200 base pairs long.

The human promoter dataset contains promoter activity (CRE fitness) measured from three leukemia-derived cell lines: JURKAT, K562, and THP1. All three are mesoderm-derived hematopoietic cell lines and share high biological similarity. Each sequence in this dataset is 250 base pairs in length. Compared to enhancer datasets that span multiple germ layers and tissue types, optimization and constraint satisfaction in the promoter dataset is more challenging due to the biological similarity between the cell lines [4].

We provide percentile statistics of normalized activity scores in Tables 4 and 5. Notably, in the THP1 cell line, even the 25th percentile activity reaches 0.49, suggesting a right-skewed distribution. This distributional bias may partially explain the increased difficulty in constraining THP1 activity, as discussed in Section 4.2.

Table 4: Percentile statistics of normalized activity scores across cell types in Human Enhancer datasets

| Cell Line | 25th Percentile | 50th Percentile | 75th Percentile | 90th Percentile |
|-----------|-----------------|-----------------|-----------------|-----------------|
| HepG2 | 0.34 | 0.36 | 0.40 | 0.45 |
| K562 | 0.34 | 0.36 | 0.40 | 0.45 |
| SK-N-SH | 0.35 | 0.37 | 0.40 | 0.45 |

Table 5: Percentile statistics of normalized activity scores across cell types in Human Promoter datasets.

| Cell Line | 25th Percentile | 50th Percentile | 75th Percentile | 90th Percentile |
|---|---|---|---|---|
| JURKAT | 0.35 | 0.38 | 0.44 | 0.54 |
| K562 | 0.23 | 0.26 | 0.32 | 0.40 |
| THP1 | 0.49 | 0.51 | 0.53 | 0.59 |

## A.5 Baselines

In this section, we provide detailed descriptions of all baseline methods compared in the main paper:

- **AdaLead** [45]: is implemented as a novelty-guided hill-climbing algorithm with mutation rate $\mu =1$ (applied as $\mu/L$ per position), recombination rate r $=0.2$, and greedy threshold $k= 0.05$ for selecting parents with fitness above $(1-k)$ of the current best. Recombination is disabled ($\rho= 0$), and each candidate is evaluated with $v= 20$ model queries.

- **Bayesian Optimization (BO)** [45]: A black-box optimization method that models the fitness function with a Gaussian process surrogate and selects new candidates by maximizing an acquisition function.

- **CMA-ES** [47]: A population-based evolutionary algorithm that adapts a multivariate Gaussian distribution over iterations. We apply CMA-ES on one-hot encoded sequence representations with a population size of 16 and initial search variance of 0.2. The number of iterations is scaled to the model query budget.

- **PEX** [48]: An evolutionary approach that prioritizes generating high-fitness variants with minimal mutations relative to the wild-type sequence. It generates candidates by applying 2 random mutations to high-fitness frontier sequences, stratified by Hamming distance from the wild-type. The frontier neighbor size is set to 5, and predictions are processed in batches of 64.

- **RegLM** [8]: An autoregressive language model trained to generate cis-regulatory elements (CREs) conditioned on cell-type-specific fitness profiles. We fine-tune the hyenadna-medium-160k-seqlen configuration (6.55M parameters) for 16 epochs, using the AdamW optimizer (learning rate = 0.0001, batch size = 1024) and cross-entropy loss, with inverse-frequency label sampling to address class imbalance.

- **TACO** [9]: A reinforcement learning method based on REINFORCE [34], which incorporates transcription factor motif rewards to guide generation toward high-fitness sequences. Its implementation follows the original paper's hyperparameter settings. The TFBS reward coefficient is set to 0.01. We use a batch size of 256 and run optimization for 100 iterations. The learning rate is fixed at 1e-4 across all datasets.

- **PPO** [35]: A widely used policy optimization algorithm that updates policies using clipped surrogate objectives to ensure stable training. We use Generalized Advantage Estimation (GAE) to compute the advantage function. Optimization is performed using the Adam optimizer with a learning rate of 1e-4. The value network shares the same pretrained HyenaDNA backbone as the policy network, with a linear head on top of the final hidden state to predict value estimates. We use a batch size of 256 and train for 100 iterations.

- **PPO-Lagrangian** [12]: A constrained variant of PPO that incorporates Lagrangian relaxation to balance main rewards and constraint satisfaction. For each reward component (i.e., each cell type), a separate value network is trained to estimate the expected return for that component. These value networks share the same architecture as in PPO. However, due to GPU memory constraints, we apply LoRA to fine-tune only the MLP and mixer layers in the HyenaDNA model. The optimizer, learning rate (1e-4), batch size (256), and total training iterations (100) are kept consistent with the PPO setup.

## A.6 Sequence Diversity

In this section, we report the full results of sequence diversity for *Ctrl-DNA* and baseline methods as shown in Figure 4.

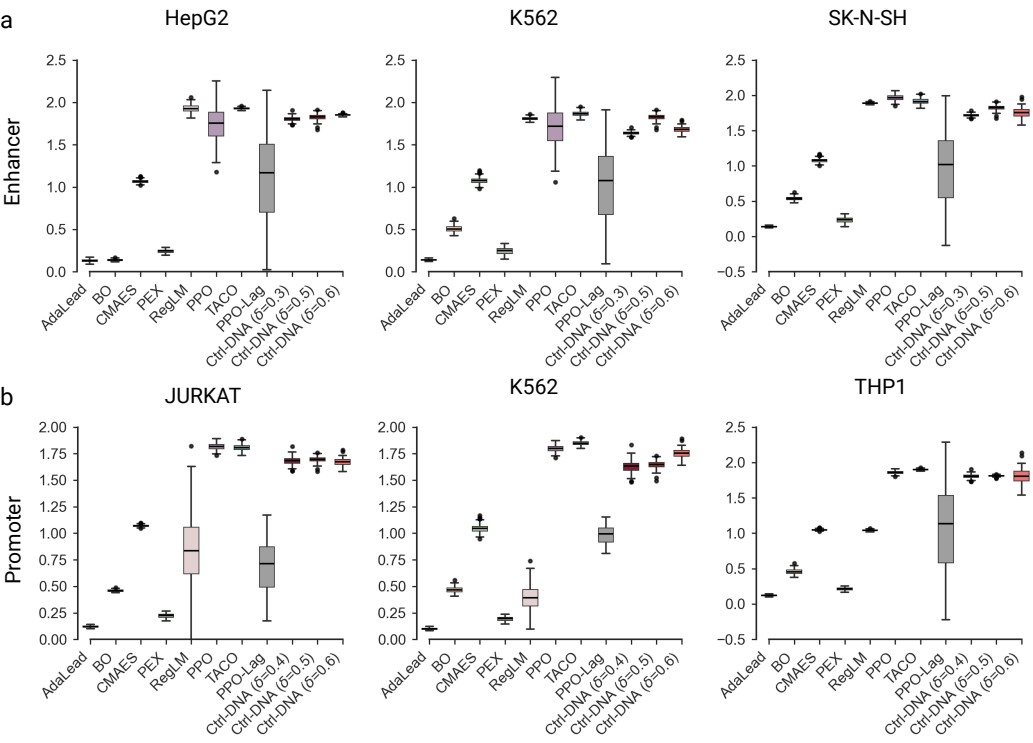

Figure 4: Sequence diversity scores for generated sequences on the human enhancer and promoter datasets. Higher values indicate greater variability among generated sequences.

## A.7 Extended Ablation Results

This section provides additional results and implementation details for the ablation experiments introduced in Section 4.4. We evaluate ablations on both the Human Enhancer and Human Promoter datasets.

**Ctrl-DNA-Log.** Following the reward-guided approach proposed in [50], we implement a log-barrier transformation of the constraint reward. Specifically, we define a log-augmented reward as

$$R_{\log}(X) = R_0(X) + \sum_{i=1}^{m} \log\left(\max\left(\delta_i - R_i(X), c_1\right)\right), \qquad (9)$$

where $R_0$ is the target reward, $R_i$ is the constraint reward for $i \geq 1$, $c$ is the threshold, and $c_1$ is a small constant for numerical stability. We then compute normalized advantages using this transformed reward:

$$A_{log}^{(j)} = \frac{R_{\log}(X_j) - \bar{R}_{\log}}{\sigma(R_{\log})}. \qquad (10)$$

We replace the mixed advantages in the original loss function (Equation 6) with $A_{\log}^{(j)}$. All other settings (e.g., surrogate loss, clipping, KL regularization) remain the same as in *Ctrl-DNA*.

Table 6: Ablation study for *Ctrl-DNA* across three target cell types in Human Enhancer datasets.

| Method | Target: HepG2 | | | | | |
|---|---|---|---|---|---|---|
| | HepG2 ↑ | K562 ↓ | SK-N-SH ↓ | $\Delta R$ | Motif Correlation | Diversity |
| Ctrl-DNA-Log | 0.24 (0.02) | 0.24 (0.03) | 0.21 (0.04) | 0.02 (0.06) | 0.16 (0.13) | 1.62 (0.08) |
| Ctrl-DNA-IPO | 0.74 (0.01) | 0.86 (0.02) | 0.83 (0.02) | -0.10 (0.02) | 0.39 (0.14) | 1.58 (0.13) |
| Ctrl-DNA ( $\lambda_{max} = 0.00$) | 0.78 (0.02) | 0.40 (0.06) | 0.33 (0.03) | 0.42 (0.02) | 0.33 (0.12) | 1.82 (0.02) |
| Ctrl-DNA ( $\lambda_{max} = 0.01$) | 0.77 (0.01) | 0.34 (0.04) | 0.30 (0.02) | 0.45 (0.02) | 0.16 (0.10) | 1.84 (0.01) |
| Ctrl-DNA ( $\lambda_{max} = 0.1$) | 0.77 (0.01) | 0.36 (0.04) | 0.31 (0.04) | 0.44 (0.03) | 0.43 (0.07) | 1.82 (0.04) |

| Method | Target: K562 | | | | | |
|---|---|---|---|---|---|---|
| | K562 ↑ | HepG2 ↓ | SK-N-SH ↓ | $\Delta R$ | Motif Correlation | Diversity |
| Ctrl-DNA-Log | 0.28 (0.01) | 0.16 (0.06) | 0.19 (0.02) | 0.11 (0.02) | 0.14 (0.07) | 1.68 (0.16) |
| Ctrl-DNA-IPO | 0.84 (0.11) | 0.65 (0.09) | 0.73 (0.19) | 0.15 (0.03) | 0.39 (0.05) | 1.08 (0.62) |
| Ctrl-DNA ($\lambda_{max} = 0.00$) | 0.93 (0.01) | 0.43 (0.01) | 0.35 (0.01) | 0.54 (0.01) | 0.50 (0.02) | 1.72 (0.02) |
| Ctrl-DNA ($\lambda_{max} = 0.01$) | 0.93 (0.01) | 0.42 (0.05) | 0.35 (0.01) | 0.54 (0.06) | 0.52 (0.03) | 1.73 (0.05) |
| Ctrl-DNA ($\lambda_{max} = 0.1$) | 0.93 (0.01) | 0.43 (0.01) | 0.35 (0.01) | 0.54 (0.01) | 0.51 (0.02) | 1.82 (0.04) |

| Method | Target: SK-N-SH | | | | | |
|---|---|---|---|---|---|---|
| | SK-N-SH ↑ | HepG2 ↓ | K562 ↓ | $\Delta R$ | Motif Correlation | Diversity |
| Ctrl-DNA-Log | 0.46 (0.07) | 0.05 (0.01) | 0.05 (0.01) | 0.41 (0.07) | 0.12 (0.02) | 1.68 (0.08) |
| Ctrl-DNA-IPO | 0.91 (0.04) | 0.87 (0.01) | 0.70 (0.01) | 0.13 (0.03) | 0.13 (0.04) | 1.66 (0.07) |
| Ctrl-DNA ($\lambda_{max} = 0.00$) | 0.83 (0.04) | 0.57 (0.11) | 0.38 (0.07) | 0.35 (0.10) | 0.35 (0.06) | 1.78 (0.03) |
| Ctrl-DNA ($\lambda_{max} = 0.01$) | 0.88 (0.01) | 0.47 (0.05) | 0.30 (0.03) | 0.49 (0.03) | 0.15 (0.02) | 1.84 (0.04) |
| Ctrl-DNA ($\lambda_{max} = 0.1$) | 0.86 (0.02) | 0.54 (0.13) | 0.44 (0.01) | 0.37 (0.11) | 0.51 (0.02) | 1.82 (0.04) |

Table 7: Ablation study for *Ctrl-DNA* across three target cell types in Human Promoter datasets.

| Method | Target: JURKAT | | | | | |
|---|---|---|---|---|---|---|
| | JURKAT ↑ | K562 ↓ | THP1 ↓ | $\Delta R$ | Motif Correlation | Diversity |
| Ctrl-DNA-Log | 0.46 (0.02) | 0.17 (0.02) | 0.45 (0.02) | 0.15 (0.01) | 0.11 (0.36) | 1.61 (0.07) |
| Ctrl-DNA-IPO | 0.55 (0.14) | 0.28 (0.18) | 0.55 (0.15) | 0.15 (0.01) | 0.31 (0.29) | 1.22 (0.38) |
| Ctrl-DNA ($\lambda_{max} = 0.00$) | 0.59 (0.11) | 0.19 (0.03) | 0.49 (0.02) | 0.25 (0.09) | 0.28 (0.36) | 1.56 (0.09) |
| Ctrl-DNA ($\lambda_{max} = 0.01$) | 0.56 (0.12) | 0.18 (0.02) | 0.49 (0.02) | 0.22 (0.10) | 0.18 (0.31) | 1.56 (0.07) |
| Ctrl-DNA ($\lambda_{max} = 0.1$) | 0.69 (0.09) | 0.38 (0.02) | 0.49 (0.01) | 0.25 (0.01) | 0.69 (0.01) | 1.69 (0.03) |

| Method | Target: K562 | | | | | |
|---|---|---|---|---|---|---|
| | K562 ↑ | JURKAT ↓ | THP1 ↓ | $\Delta R$ | Motif Correlation | Diversity |
| Ctrl-DNA-Log | 0.26 (0.04) | 0.29 (0.03) | 0.44 (0.03) | -0.11 (0.06) | 0.60 (0.01) | 1.49 (0.11) |
| Ctrl-DNA-IPO | 0.59 (0.02) | 0.59 (0.06) | 0.69 (0.09) | -0.11 (0.06) | 0.57 (0.05) | 1.65 (0.14) |
| Ctrl-DNA ($\lambda_{max} = 0.00$) | 0.61 (0.04) | 0.49 (0.03) | 0.49 (0.01) | 0.11 (0.03) | 0.61 (0.16) | 1.63 (0.06) |
| Ctrl-DNA ($\lambda_{max} = 0.01$) | 0.58 (0.04) | 0.52 (0.03) | 0.49 (0.01) | 0.08 (0.02) | 0.67 (0.05) | 1.62 (0.11) |
| Ctrl-DNA ($\lambda_{max} = 0.1$) | 0.58 (0.03) | 0.43 (0.06) | 0.50 (0.01) | 0.12 (0.02) | 0.75 (0.06) | 1.64 (0.04) |

| Method | Target: THP1 | | | | | |
|---|---|---|---|---|---|---|
| | THP1 ↑ | JURKAT ↓ | K562 ↓ | $\Delta R$ | Motif Correlation | Diversity |
| Ctrl-DNA-Log | 0.51 (0.01) | 0.10 (0.04) | 0.16 (0.01) | 0.38 (0.02) | 0.42 (0.05) | 1.50 (0.03) |
| Ctrl-DNA-IPO | 0.88 (0.04) | 0.59 (0.06) | 0.42 (0.11) | 0.38 (0.01) | 0.57 (0.05) | 1.77 (0.06) |
| Ctrl-DNA ($\lambda_{max} = 0.00$) | 0.92 (0.01) | 0.51 (0.02) | 0.33 (0.05) | 0.50 (0.03) | 0.23 (0.07) | 1.82 (0.01) |
| Ctrl-DNA ($\lambda_{max} = 0.01$) | 0.92 (0.01) | 0.50 (0.01) | 0.32 (0.01) | 0.51 (0.02) | 0.23 (0.02) | 1.82 (0.01) |
| Ctrl-DNA ($\lambda_{max} = 0.1$) | 0.92 (0.01) | 0.51 (0.01) | 0.31 (0.03) | 0.51 (0.01) | 0.25 (0.04) | 1.81 (0.01) |

**Ctrl-DNA-IPO.** Based on Interior-point Policy Optimization (IPO) [49], we incorporate the log-barrier directly into the optimization objective. The surrogate loss becomes:

$$\mathcal{L}^{\text{IPO}}(\theta) = \frac{1}{B} \sum_{j=1}^{B} \sum_{i=1}^{T} \min \left\{ \rho_i^{(j)} A_0^{(j)}, \ \text{clip}_\epsilon(\rho_i^{(j)}) A_0^{(j)} \right\} - \beta \cdot \text{KL}(\pi_\theta \,||\, \pi_{\text{old}}) - \sum_{i=1}^{m} \phi\left( \hat{J}_i^{\pi_\theta} \right), \quad (11)$$

where the log-barrier penalty is defined as

$$\phi(\hat{J}_i^{\pi_\theta}) = \frac{1}{t} \log\left( \delta_i - \hat{J}_i^{\pi_\theta} \right), \quad (12)$$

with $t > 0$ controlling the sharpness of the approximation to the indicator function. A larger $t$ yields a tighter barrier. We set $t = 50$ in our experiments. Note that, unlike the main *Ctrl-DNA* method, this variant does not compute mixed advantages as in Equation 5. Instead, we compute advantages using only the target reward:

$$A_0^{(j)} = \frac{R_0(X_j) - \bar{R}_0}{\sigma(R_0)}. \quad (13)$$

## A.8  Differential Expression Optimization

To assess whether a scalarized objective suffices for cell-type-specific design, we replace the original constrained reward in all optimizers with the differential expression (DE) objective [4]. Concretely, for each target cell type $t$ and set of off-target cell types $\mathcal{O}$, we optimize a single reward of the form

$$\text{DE}(x) = R_0(x) - \frac{1}{m} \sum_{i=1}^{m} \{R_i(x)\},$$

where $R_0(x)$ are the cell activity of sequence x in the target cell type. We evaluate TACO, AdaLead, PEX, and Ctrl-DNA (ours) on the human enhancer tasks by substituting their training rewards with $\text{DE}(x)$; we also include a Ctrl-DNA-DE variant that uses DE as its reward during policy updates. The rest of the training and evaluation protocol is unchanged from the main experiments.

We additionally include Simulated Annealing (SA) for probabilistic exploration and Fast SeqProp (FSP) for gradient-based refinement [38]. These two methods optimize *MinGap*, which is a specific scalarization similar to DE. The DE-based evaluations are therefore directly comparable to their standard objective.

As reported in Table 8, DE-based baseline variants consistently perform worse than Ctrl-DNA, even in terms of DE values. This is because using a DE reward imposes a fixed trade-off between target and off-target expression, which often results in suboptimal solutions when the objectives conflict. This fixed weighting lacks the flexibility to adjust the balance between objectives throughout training [52]. In contrast, our Lagrangian approach allows for a more effective coordination of competing goals, resulting in better performance.

Figure 2 further shows that Ctrl-DNA supports variable off-target thresholds (e.g., 0.3, 0.5, 0.6), allowing users to tailor optimization to specific biological requirements. Once the constraint is satisfied, the model focuses on maximizing on-target activity within the feasible region. In contrast, DE optimization can lead the model to suppress off-target activity, even at the cost of reducing on-target activation. For example, when optimizing for SK-N-SH , Ctrl-DNA-DE tends to suppress off-target expression at the expense of decreasing SK-N-SH cell activity.

## A.9  Reward Model Generalization

We optimized and evaluated sequences using the same pretrained reward model from regLM [4], trained on promoter data with chromosomes 7, 13, 21, and 22 held out. To test robustness to *reward model shift* (and potential reward hacking), we introduced two evaluation settings: (1) optimization uses the regLM reward model, while evaluation uses a separate "full" reward model trained on the entire dataset (including all chromosomes); (2) optimization uses a reward model trained on a random 80% subset of the full data, while evaluation again uses the same full reward model as in Setting 1.

Table 8: DE optimization study for *Ctrl-DNA* across three target cell types in Human Enhancer datasets.

| Method | Target: HepG2 | | | |
| --- | --- | --- | --- | --- |
| | HepG2 ↑ | K562 ↓ | SK-N-SH ↓ | $\Delta R$ |
| Ctrl-DNA | 0.77 (0.01) | 0.36 (0.04) | 0.31 (0.04) | **0.44 (0.03)** |
| Ctrl-DNA-DE | 0.67 (0.02) | 0.47 (0.02) | 0.37 (0.03) | 0.25 (0.02) |
| TACO-DE | 0.58 (0.05) | 0.46 (0.04) | 0.39 (0.01) | 0.16 (0.03) |
| AdaLead-DE | 0.45 (0.03) | 0.46 (0.05) | 0.43 (0.13) | -0.01 (0.02) |
| Pex-DE | 0.67 (0.06) | 0.57 (0.03) | 0.61 (0.04) | 0.06 (0.03) |
| FSP | 0.62 (0.03) | 0.36 (0.02) | 0.36 (0.04) | 0.26 (0.03) |
| SA | 0.61 (0.01) | 0.33 (0.01) | 0.33 (0.01) | 0.28 (0.03) |

| Method | Target: K562 | | | |
| --- | --- | --- | --- | --- |
| | K562 ↑ | HepG2 ↓ | SK-N-SH ↓ | $\Delta R$ |
| Ctrl-DNA | 0.93 (0.01) | 0.43 (0.01) | 0.35 (0.01) | **0.54 (0.01)** |
| Ctrl-DNA-DE | 0.87 (0.04) | 0.43 (0.01) | 0.35 (0.02) | 0.48 (0.03) |
| TACO-DE | 0.80 (0.01) | 0.48 (0.01) | 0.49 (0.02) | 0.50 (0.01) |
| AdaLead-DE | 0.54 (0.16) | 0.49 (0.12) | 0.49 (0.15) | 0.05 (0.03) |
| Pex-DE | 0.81 (0.04) | 0.60 (0.06) | 0.61 (0.08) | 0.18 (0.07) |
| FSP | 0.67 (0.07) | 0.35 (0.02) | 0.32 (0.04) | 0.33 (0.04) |
| SA | 0.67 (0.07) | 0.34 (0.01) | 0.33 (0.01) | 0.33 (0.04) |

| Method | Target: SK-N-SH | | | |
| --- | --- | --- | --- | --- |
| | SK-N-SH ↑ | HepG2 ↓ | K562 ↓ | $\Delta R$ |
| Ctrl-DNA | 0.86 (0.02) | 0.54 (0.13) | 0.44 (0.01) | **0.37 (0.11)** |
| Ctrl-DNA-DE | 0.68 (0.02) | 0.26 (0.02) | 0.28 (0.04) | **0.41 (0.02)** |
| TACO-DE | 0.54 (0.02) | 0.42 (0.04) | 0.42 (0.09) | 0.12 (0.01) |
| AdaLead-DE | 0.45 (0.03) | 0.43 (0.03) | 0.45 (0.05) | 0.01 (0.02) |
| Pex-DE | 0.77 (0.03) | 0.64 (0.09) | 0.67 (0.09) | 0.09 (0.02) |
| FSP | 0.54 (0.03) | 0.39 (0.02) | 0.36 (0.01) | 0.16 (0.02) |
| SA | 0.54 (0.04) | 0.37 (0.05) | 0.35 (0.04) | 0.18 (0.05) |

In all cases we use the human enhancer dataset targeting HepG2-specific activity, and the evaluation protocol matches the main experiments.

As shown in Table 9, both settings have trends and values consistent with the main results on expression predictions, indicating that the observed gains are not artifacts of overfitting to the training reward. In particular, evaluating under a distinct, stronger reward model (Settings 1–2) preserves the relative performance of Ctrl-DNA, suggesting robustness to evaluator shift and minimal reward hacking.

Table 9: Reward model generalization evaluation on the human enhancer task (HepG2 target).

| Method | HepG2 ↑ | K562 ↓ | SK-N-SH ↓ | $\Delta R$ ↑ |
| --- | --- | --- | --- | --- |
| Ctrl-DNA | 0.77 (0.01) | 0.36 (0.04) | 0.31 (0.04) | 0.44 (0.03) |
| Setting 1 | 0.76 (0.02) | 0.35 (0.03) | 0.35 (0.02) | 0.40 (0.04) |
| Setting 2 | 0.82 (0.04) | 0.37 (0.08) | 0.36 (0.02) | 0.45 (0.01) |

## A.10 Extended TFBS Reward Analysis

A potential concern is the use of motif-frequency correlation both as a regularizer during training and as an evaluation metric. To address concerns of circularity, we conducted an additional analysis using distinct reference sets for evaluation, created by splitting sequences at the 90th and 50th percentiles based on their on and off target activities. The motif frequency correlation results are evaluated by using the new reference sets (Table 10). Although this filtering significantly reduced the dataset size (leaving only 18 sequences for JURKAT, 4 for K562, and 7 for THP1 out of the original 12,335), Ctrl-DNA continued to demonstrate strong performance, substantially outperforming baseline methods across all tested cell lines with correlations of 0.37 for JURKAT, 0.52 for K562, and 0.60 for THP1.

Table 10: Motif-frequency correlation under 90/50 percentile selection.

| Cell Type | Ctrl-DNA | AdaLead | BO | CMA-ES | PEX | PPO-Lag | PPO | TACO |
|---|---|---|---|---|---|---|---|---|
| JURKAT | **0.37 (0.02)** | 0.23 (0.10) | 0.10 (0.17) | 0.19 (0.06) | 0.34 (0.02) | 0.27 (0.15) | 0.32 (0.12) | 0.39 (0.09) |
| K562 | **0.52 (0.11)** | **0.54 (0.23)** | 0.13 (0.25) | 0.14 (0.05) | 0.31 (0.02) | 0.18 (0.18) | 0.31 (0.13) | 0.21 (0.14) |
| THP1 | **0.60 (0.02)** | 0.16 (0.13) | 0.06 (0.08) | 0.06 (0.04) | 0.04 (0.02) | -0.02 (0.07) | 0.37 (0.04) | 0.33 (0.01) |

Beyond circularity, we emphasize that the TFBS reward is *global*: we first compute a motif-frequency vector from a reference set of real sequences, then use the Pearson correlation between this reference vector and the motif frequencies of generated sequences. This encourages distributional alignment of motif usage rather than direct motif matching, in contrast to methods such as TACO. Despite this coarse-grained signal, Ctrl-DNA still captures cell-type–specific motifs (e.g., HNF4A in HepG2, GATA1 in K562), indicating that specificity emerges from optimizing the global reward in conjunction with explicit off-target constraints (see Fig. 3a).

Regarding threshold choice, we follow prior work [8, 9] and adopt percentile-based selection (on-target $\geq$ 50th; off-target $\leq$ 50th) to reduce sensitivity to cross-cell-type scaling and normalization. While a stricter 90th-percentile cutoff for on-target activity can enrich for highly specific sequences, it yields too few samples for reliable TFBS enrichment estimates (e.g., 18/4/7 for JURKAT/K562/THP1 in promoters; none across all three cell types in the human enhancer set among $\sim$660k candidates). To further guard against overfitting to spurious motif correlations, Ctrl-DNA employs learnable, upper-bounded weights for each TFBS regularization term (Section 3.3), which regularizes the influence of any single motif channel.

## A.11 Effect of Pretraining on Optimization

We use pretrained HyenaDNA backbones without task-specific fine-tuning prior to reinforcement learning. To quantify the impact of pretraining, we ablate the model initialization on the HepG2 enhancer task by comparing a pretrained backbone to a randomly initialized one. Training and evaluation protocols match the main setup. We do not employ fine-tuned backbones in the main results because prior work [6] and our early trials showed minimal benefits over pretraining alone.

Table 11: Pretraining ablation on the HepG2 enhancer task.

| Method | HepG2 $\uparrow$ | K562 $\downarrow$ | SK-N-SH $\downarrow$ | $\Delta R \uparrow$ |
|---|---|---|---|---|
| Pretrained | 0.77 (0.01) | 0.36 (0.04) | 0.31 (0.04) | 0.44 (0.03) |
| Random Init. | 0.71 (0.01) | 0.49 (0.03) | 0.42 (0.02) | 0.25 (0.01) |

Pretraining materially improves target activation while reducing off-target expression, yielding a substantially higher $\Delta R$ (Table 11). Notably, TACO also uses a pretrained HyenaDNA backbone (Fig. 2) yet underperforms Ctrl-DNA, indicating that pretraining helps but does not explain the performance gap; the optimization framework remains the dominant factor.

## A.12 Additional Experiments

We extend our human enhancer experiments with two additional baselines. Specifically, we add DRAKES, a diffusion-based generator for regulatory sequences [53], and CbAS, a model-based design method that conditions a search distribution on meeting a property event via importance-weighted maximum likelihood [54]. Results are reported in Table 12. Across all three target cell types (HepG2, K562, and SK-N-SH), Ctrl-DNA consistently achieves the highest target-specific activity and the largest $\Delta R$, indicating superior ability to enhance the target cell type while suppressing off-target expression. For example, when targeting HepG2, Ctrl-DNA improves $\Delta R$ to 0.44 $\pm$ 0.03, compared to near-zero or negative shifts for CbAS and DRAKES, demonstrating effective specificity control. Similar gains are observed for K562 ($\Delta R = 0.54 \pm 0.01$) and SK-N-SH ($\Delta R = 0.37 \pm 0.11$). These results highlight the advantage of Ctrl-DNA in handling multiple constraints compared to diffusion and model-based methods optimized for single objectives.

Table 12: Additional experiments for *Ctrl-DNA* across three target cell types in Human Enhancer datasets.

| Method | Target: HepG2 | | | |
| --- | --- | --- | --- | --- |
| | HepG2 ↑ | K562 ↓ | SK-N-SH ↓ | $\Delta R$ |
| Ctrl-DNA | 0.77 (0.01) | 0.36 (0.04) | 0.31 (0.04) | **0.44 (0.03)** |
| CbAS | 0.74 (0.03) | 0.79 (0.04) | 0.73 (0.03) | -0.02 (0.02) |
| DRAKES | 0.68 (0.004) | 0.78 (0.007) | 0.71 (0.007) | -0.07 (0.005) |

| Method | Target: K562 | | | |
| --- | --- | --- | --- | --- |
| | K562 ↑ | HepG2 ↓ | SK-N-SH ↓ | $\Delta R$ |
| Ctrl-DNA | 0.93 (0.01) | 0.43 (0.01) | 0.35 (0.01) | **0.54 (0.01)** |
| CbAS | 0.90 (0.01) | 0.66 (0.02) | 0.69 (0.01) | 0.23 (0.18) |
| DRAKES | 0.83 (0.003) | 0.69 (0.002) | 0.80 (0.001) | 0.08 (0.002) |

| Method | Target: SK-N-SH | | | |
| --- | --- | --- | --- | --- |
| | SK-N-SH ↑ | HepG2 ↓ | K562 ↓ | $\Delta R$ |
| Ctrl-DNA | 0.86 (0.02) | 0.54 (0.13) | 0.44 (0.01) | **0.37 (0.11)** |
| CbAS | 0.81 (0.05) | 0.68 (0.03) | 0.81 (0.06) | 0.06 (0.01) |
| DRAKES | 0.79 (0.001) | 0.80 (0.003) | 0.69 (0.002) | 0.05 (0.002) |

