# OpenReview forum: "Ctrl-DNA: Controllable Cell-Type-Specific Regulatory DNA Design via Constrained RL"
_NeurIPS.cc/2025/Conference — NeurIPS 2025 spotlight_

### Official Review · Reviewer_t2FQ · 2025-07-02

**Clarity:** 3
**Significance:** 3
**Originality:** 3
**Rating:** 5
**Confidence:** 5

**Summary:**

The paper presents regCon, a framework that combines an autoregressive DNA language model with constrained RL, resulting in the generation of cell type-specific enhancers or promoters. The method limits the activity of the generated sequences in off-target cell types while maximizing activity in the on-target cell type. The generated sequences are predicted to be specific, diverse, and contain biologically reasonable transcription factor binding motifs.

**Questions:**

1. A comparison to one or more diffusion modeling approaches would be useful, as several methods have been proposed recently to optimize diffusion models for cell type-specific DNA generation. These would be a conceptually similar benchmark for regCon.

2. Motif frequency correlation is used as a regularization term and also as an evaluation metric in table 1. Is this term the same, and if so, is this circular?

3. How is the TFBS frequency used for regularization computed? Is it based on the 50th percentile or 90th percentile of sequences? Would it not be better to set this threshold to an absolute value based on the target level of cell type-specificity, or to always use the 90th percentile? The choice of 50th percentile (Line 217) was confusing as the aim of the paper is to generate sequences with extremely high specificity.

**Ethical Concerns:**

["NO or VERY MINOR ethics concerns only"]

**Final Justification:**

I am satisfied with the revisions (addition of a diffusion model baseline and clarification of the method). I have increased my score slightly and have also increased my confidence.

**Limitations:**

yes

**Quality:**

4

**Strengths And Weaknesses:**

Quality: The paper stands out in the quality of evaluation. Particularly, the authors compare their method not only to similar methods based on autoregressive LMs, but to a wide range of different approaches. They also evaluate the model on cell types that are very similar to each other, which has not typically been done in previous studies.

Clarity: Greater clarity on the computation and use of TFBS correlation would be appreciated.

Significance: This method addresses a significant problem in the field, that of generating extremely cell type specific regulatory elements. This problem is critical for many industrial and therapeutic applications and such regulatory elements are extremely rare in nature.

Originality: The method is quite original in the field. Previous work on using autoregressive language models for DNA sequence generation was limited in utility as it did not optimize the generated sequences to be extremely cell type-specific, which is a highly desirable property for gene therapy and other applications. This method effectively addresses that limitation, potentially making autoregressive language models more valuable to design sequences outside the naturally occurring range of functionality.

---

> ### Author Rebuttal · Authors · 2025-07-30
>
> We appreciate the reviewer’s recognition of our method’s originality, comprehensive evaluation, and potential for designing highly cell-type-specific regulatory elements. We also thank the reviewer for the constructive feedback. Please find our reponse below regarding the added diffusion baseline experiemnts and clarification on the TFBS frequency reward.
>
> ### **[Q1]**
>
> Thank you for this valuable suggestion. To address your comment, we performed an additional benchmark comparison with DRAKES[1], a recently proposed diffusion based method optimized using RL that has demonstrated success in human enhancer generation for the HepG2 cell line. Specifically, we used their publicly available checkpoint fine tuned on HepG2 and followed their experimental setup to further fine tune the model separately for K562 and SK-N-SH cell lines. It is important to note that DRAKES optimizes only a single objective, which is maximizing the expression in one target cell type. As shown in the newly added benchmark results, regCon consistently outperforms DRAKES by achieving higher expression levels in the target cell lines (0.77 versus 0.68 for HepG2, 0.93 versus 0.83 for K562, and 0.86 versus 0.79 for SK-N-SH) and significantly lower off target expressions. These results highlight the advantage of regCon in handling multiple constraints compared to diffusion based methods optimized for single objectives.
>
> | **Method**           | **HepG2 ↑**     | **K562 ↓**     | **SK-N-SH ↓** | **ΔR**        |
> |----------------------|-----------------|----------------|----------------|----------------|
> | regCon               | 0.77 (0.01)     | 0.36 (0.04)    | 0.31 (0.04)    | 0.44 (0.03)    |
> | DRAKES               | 0.68 (0.004)     | 0.78 (0.007)    | 0.71 (0.007)    | -0.07 (0.005)|
>
> | **Method**           | **K562 ↑**     | **HepG2↓**     | **SK-N-SH ↓** | **ΔR**        |
> |----------------------|-----------------|----------------|----------------|----------------|
> | regCon               | 0.93 (0.004)     |  0.43 (0.01)   | 0.35 (0.007)    |    0.54 (0.01) |
> | DRAKES               | 0.83 (0.003)     | 0.69 (0.002)    | 0.80 (0.001)    | 0.08 (0.002)|
>
> | **Method**           | **SK-N-SH ↑**     | **K562 ↓**     | **HepG2↓** | **ΔR**        |
> |----------------------|-----------------|----------------|----------------|----------------|
> | regCon               | 0.86 (0.02)     | 0.44 (0.01)    | 0.54 (0.13)    |    0.37 (0.11)|
> | DRAKES               | 0.79 (0.001)     | 0.80 (0.003)    | 0.69 (0.002)    | 0.05 (0.002)|
>
> ### **[Q2]**
>
>
> Thank you for raising this important question. We acknowledge the potential limitation arising from the use of motif frequency correlation as both a regularization term and an evaluation metric. To address concerns of circularity, we conducted an additional analysis using distinct reference sets for evaluation, created by splitting sequences at the 90th and 50th percentiles based on their on and off target activities. Although this filtering significantly reduced the dataset size (leaving only 18 sequences for JURKAT, 4 for K562, and 7 for THP1 out of the original 12,335), regCon continued to demonstrate strong performance, substantially outperforming baseline methods across all tested cell lines with correlations of 0.37 for JURKAT, 0.52 for K562, and 0.60 for THP1.
>
> | Cell Type | *regCon*       | AdaLead     | BO          | CMAES       | PEX         | PPO-Lag        | PPO         | TACO        |
> |-----------|------------|-------------|-------------|-------------|-------------|-------------|-------------|-------------|
> | JURKAT    | ***0.37 (0.02)***| 0.23 (0.10) | 0.10 (0.17) | 0.19 (0.06) | 0.34 (0.02) | 0.27 (0.15) | 0.32 (0.12) | ***0.39 (0.09)*** |
> | K562      | ***0.52 (0.11)***| ***0.54 (0.23)*** | 0.13 (0.25) | 0.14 (0.05) | 0.31 (0.02) | 0.18 (0.18) | 0.31 (0.13) | 0.21 (0.14) |
> | THP1      | ***0.60 (0.02)***| 0.16 (0.13) | 0.06 (0.08) | 0.06 (0.04) | 0.04 (0.02) | -0.02(0.07) | 0.37 （0.04）| 0.33（0.01）|
>
> We also want to emphasize that using motif frequency correlation as a reward promotes generalizability, as shown in Figure 3a of our manuscript. Specifically, the TFBS reward is computed by first deriving a motif frequency vector from a reference set of real DNA sequences, then calculating the Pearson correlation between this reference vector and the motif frequencies in generated sequences. This encourages the model to capture global frequency patterns rather than explicitly match individual motifs, in contrast to prior methods like TACO. Despite this global formulation, regCon still successfully generates cell type–specific motifs such as HNF4A for HepG2 and GATA1 for K562, demonstrating that it acquires this ability purely through optimization on a coarse-grained reward. This highlights the robustness and generalizability of regCon’s optimization strategy.
>
>
> ### **[Q3]**
> Thank you for the insightful comment. In our experiments, we selected the sequences with activity above the 50th percentile in the target cell type, and off-target sequences as those with activity below the 50th percentile in non-target cell types. We then computed each motif’s occurrence frequency from these selected sequences. The motifs were selected following the protocol in [2].
>
> This percentile-based thresholding, as adopted from prior work [2,3], helps mitigate differences in expression scaling across cell types and avoids reliance on absolute activity values, which may vary due to experimental noise or normalization schemes.
>
> While we agree that using a more stringent threshold, such as the 90th percentile for on-target activity, could better capture highly specific sequences, doing so significantly reduces the number of selected sequences. For instance, in the human promoter dataset, using a 90th/50th percentile split for on/off-target activity results in only 18 sequences for JURKAT, 4 for K562, and 7 for THP1, out of a total of 12,335. In the human enhancer dataset, no sequences satisfy such criteria across all three cell types among around 660,000 candidates. These sample sizes are insufficient to support reliable transcription factor binding site (TFBS) motif enrichment analysis. Therefore, we opted to use a relaxed threshold to ensure sufficient statistical power and robustness in estimating TFBS frequencies.
>
> Importantly, to mitigate potential overfitting to spurious motif correlations, we use a learnable weight for each TFBS regularization term and enforce an upper bound on these weights, as described in Line 193.
>
> That said, we agree with the reviewer that using a 90th percentile cutoff would be preferable in scenarios where sufficient data is available, and we consider this a promising direction for future work.
>
> Thank you again for your valuable suggestions, which have helped improve the clarity and contribution of our work. Please let us know if you have any more questions.
> **Reference:**
>
> [1] Wang, C., Uehara, M., He, Y., Wang, A., Biancalani, T., Lal, A., ... & Regev, A. (2024). Fine-tuning discrete diffusion models via reward optimization with applications to dna and protein design. arXiv preprint arXiv:2410.13643.
>
> [2] Lal, A., Garfield, D., Biancalani, T., & Eraslan, G. (2024). Designing realistic regulatory DNA with autoregressive language models. Genome Research, 34(9), 1411-1420.
>
> [3] Yang, Z., Su, B., Cao, C., & Wen, J. R. Regulatory DNA Sequence Design with Reinforcement Learning. In The Thirteenth International Conference on Learning Representations.

---

> ### Comment · Reviewer_t2FQ · 2025-08-01
>
> I am satisfied with the authors' response. The comparison to DRAKES is a useful addition to the paper. I agree that the amount of data available limits the percentile cutoff that can be applied. It would be good to mention in the paper that this is the reason for choosing 50th percentile as the cutoff.
>
> I think this method is especially useful to the field because of the explicit constraints enforced on the off-target cell activity. Current methods based on simple DE optimization are difficult to use for therapeutics design since they can easily result in excessive off-target activity (leading to potential side effects) or insufficient on-target activity (lack of efficacy). The evaluation convincingly shows that regCon provides a practical advantage in this area.

---

> > ### Author Response · Authors · 2025-08-04
> >
> > Thank you for your thoughtful feedback and encouraging comments. We will update the manuscript to clarify the rationale behind the 50th percentile cutoff and appreciate your recognition of regCon’s practical value in therapeutic design.

---

### Official Review · Reviewer_saSR · 2025-07-02

**Clarity:** 2
**Significance:** 3
**Originality:** 3
**Rating:** 5
**Confidence:** 3

**Summary:**

This work proposes `regCon`, a novel RL method for designing cell-type-specific CREs. `regCon` formulates this task as a constrained optimization problem, aiming to maximize gene expression activity in a target cell type while satisfying user-defined thresholds for off-target cells. This framework fine-tunes a pre-trained genomic language model using a primal-dual approach based on Lagrangian relaxation. Following the recently proposed GRPO framework, it computes advantages directly from batch-normalized rewards to guide policy updates. Furthermore, to enhance biological plausibility, `regCon` incorporates a regularization term based on the correlation of TFBS frequencies between generated and real sequences. Evaluations on human promoter and enhancer datasets show that `regCon` consistently outperforms existing baselines. It successfully generates regulatory sequences with high activity in the target cell type while effectively suppressing off-target activity, achieving controllable cell-type specificity.

**Questions:**

- Line 474 states that training was conducted for 100 epochs. Does 'epoch' in this context mean 'iteration'? The term 'epoch' is less common in standard reinforcement learning terminology.
- Why was the problem formulated as a constrained optimization problem? For instance, Reddy et al. [1] define the difference of expression (DE) as a single objective. The authors should clarify the motivation for their chosen formulation.
- Are most of the implemented baselines single-objective optimizers (i.e., without explicit constraints)? It seems that if the objective were defined as DE, other methods could also be easily adapted to a multi-objective setting.
- Is the core difference between PPO-Lagrangian and `regCon` the use of a GRPO-like algorithm for advantage estimation? Why is the performance gap between them so large, and what is the primary source of `regCon`'s improvement over PPO-Lagrangian?
- How were the penalty coefficients ($\lambda$) initialized at the beginning of training?

[1] Reddy, Aniketh Janardhan, et al. "Designing cell-type-specific promoter sequences using conservative model-based optimization." Advances in Neural Information Processing Systems 37 (2024): 93033-93059.

**Ethical Concerns:**

["NO or VERY MINOR ethics concerns only"]

**Final Justification:**

The authors have addressed my two main concerns in their rebuttal: 1) the rationale for reward model selection, and 2) the respective contributions of the constrained RL formulation and GRPO. Therefore, I am raising my rating from 4 to 5.

**Limitations:**

Yes

**Paper Formatting Concerns:**

- The markers in Figure 2 could be improved. The colors and shapes for many of the methods are not easily distinguishable.

**Quality:**

3

**Strengths And Weaknesses:**

Strengths
- The paper is well-written and easy to follow.
- The problem of designing cell-type-specific CREs is of significant practical importance. The authors' choice to address this using a multi-objective optimization framework is a valuable contribution.
- The proposed method shows a significant performance improvement over the baselines.

Weaknesses
- The specific training process for the reward model and details of data partitioning should be more clearly specified in the paper. This includes, but is not limited to, the data splitting principles and the reward model's training strategy.
- Is the same reward model used for both guiding the optimization and for the final evaluation? This setup could potentially lead to reward hacking.
- The paper lacks an ablation study on the contribution of the pre-trained model. Results from a randomly initialized HyenaDNA and from the base, un-fine-tuned HyenaDNA would be valuable additions.
- The description of the baseline implementations is insufficient. While Appendix C provides a high-level overview of each method, it omits critical implementation details.

---

> ### Author Rebuttal · Authors · 2025-07-30
>
> We thank the reviewer for their thoughtful feedback and voting to accept our paper. We address each point individually below. “W/Q” numbers the weakness or question, followed by our response.
>
> ### **[w1]**
> We follow the training protocols described in [1] for our reward models. For the human enhancer task, we directly adopted the pretrained model weights from [1]. For human promoters, we trained a separate reward model using the data split from [2] (70% train, 20% validation, 10% test). Training was conducted using the AdamW optimizer with a learning rate of 1e-4 and MSE loss, over 20 epochs. The checkpoint with the best validation performance was used for evaluation.
>
> All reward models are based on an Enformer architecture [3], combining convolutional and Transformer layers, which has shown strong performance in DNA regulatory prediction tasks. Both the enhancer and promoter models share the same architecture: 11 layers with a hidden dimension of 1536. We will add these details in the revised appendix.
> ### **[w2]**
> Our method optimized and evaluated sequences using the same pretrained reward model from regLM [4], trained on promoter data excluding chromosomes 7, 13, 21, and 22. To assess potential reward hacking, we introduced two additional settings:
>
> Setting 1: Optimization used the regLM reward model; evaluation used a separate full reward model trained on the full dataset (including all chromosomes).
>
> Setting 2: Optimization used a reward model trained on a random 80% subset of the full data; evaluation used the same full reward model from Setting 1.
>
> | **Method**           | **HepG2 ↑**     | **K562 ↓**     | **SK-N-SH ↓** | **ΔR**        |
> |----------------------|-----------------|----------------|----------------|----------------|
> | regCon               | 0.77 (0.01)     | 0.36 (0.04)    | 0.31 (0.04)    | 0.44 (0.03)    |
> | Setting 1            | 0.76 (0.02)     | 0.35 (0.03)    | 0.35 (0.02)    | 0.40 (0.04)    |
> | Setting 2            | 0.82 (0.04)     | 0.37 (0.08)    | 0.36 (0.02)    | 0.45 (0.01)    |
>
>
> In both settings, we conducted experiments on the human enhancer dataset, targeting HepG2 cell-specific activity. We observed consistent performance trends across all metrics (target expression, off-target suppression, motif enrichment, and diversity), comparable to our main results.
>
> These findings suggest that *regCon* generalizes well, and that the observed performance gains are not an artifact of overfitting to the reward model used during optimization.
>
>
> ### **[W3]**
>
> We clarify that our experiments use pretrained HyenaDNA models without task-specific fine-tuning prior to reinforcement learning. To assess the role of pretraining, we compared pretrained and randomly initialized models on HepG2 enhancer optimization. We did not use fine-tuned models in our main experiments, as prior work [6] and our own early results both showed minimal performance differences between pretrained and fine-tuned models.
>
> | Method               | HepG2 ↑         | K562 ↓         | SK-N-SH ↓      | ΔR            |
> |----------------------|-----------------|----------------|----------------|----------------|
> | Pretrained           | 0.77 (0.01)     | 0.36 (0.04)    | 0.31 (0.04)    | 0.44 (0.03)    |
> | Random Init.         | 0.71 (0.01)     | 0.49 (0.03)    | 0.42 (0.02)    | 0.25 (0.01)    |
>
> We found that the randomly initialized model underperformed the pretrained model. Notably, as shown in Figure 2 in our manuscript, TACO also leverages a pretrained HyenaDNA model but still underperforms our method, indicating that while pretraining offers improvements, our framework provides more effective optimization.
>
> ### **[W4]**
> Due to the rebuttal space constraints, we are unable to include full baseline implementation details here, but we will provide them in the revised appendix.
> ### **[Q1]**:
> Thank you for pointing this out. In our implementation, we use the term "epoch" to refer to one full pass over the collected batch of trajectories. We agree that "iteration" would be a more appropriate term and will revise it in the manuscript.
> ### **[Q2 & 3]**:
> To investigate scalarized objectives, we ran additional experiments comparing regCon and the top three baselines (TACO, AdaLead, and PEX), each adapted to optimize differential expression (DE). We also tested a variant of regCon (regCon-DE) trained with DE as the reward.
>
> Our method offers two main advantages over DE objectives:
>
> - Explicit constraint control:
> As shown in Figure 2 in our paper, regCon supports variable off-target thresholds (e.g., 0.3, 0.5, 0.6), allowing users to tailor optimization to specific biological requirements. Once the constraint is satisfied, the model focuses on maximizing on-target activity within the feasible region. In contrast, DE optimization can lead the model to suppress off-target activity, even at the cost of reducing on-target activation. For example, when optimizing for SK-N-SH (Table 2), regCon-DE tends to suppress off-target expression at the expense of decreasing SK-N-SH cell activity.
>
> - Improved DE optimization:
> Most of the baselines are single-objective optimizers. Although they can be adapted to use DE as a reward, our results (Tables 1 and 2) show that these DE-based variants consistently perform worse than regCon, even in terms of DE values. This is because using a DE reward imposes a fixed trade-off between target and off-target expression, which often results in suboptimal solutions when the objectives conflict. This fixed weighting lacks the flexibility to adjust the balance between objectives throughout training [7]. In contrast, our Lagrangian approach allows for a more effective coordination of competing goals, resulting in better performance.
>
> Table 1
> | **Method**           | **HepG2 ↑**     | **K562 ↓**     | **SK-N-SH ↓** | **ΔR**        |
> |----------------------|-----------------|----------------|----------------|----------------|
> | regCon               | 0.77 (0.01)     | 0.36 (0.04)    | 0.31 (0.04)    | 0.44 (0.03)    |
> | regCon-DE            | 0.67 (0.02)     | 0.47 (0.02)    | 0.37 (0.03)    | 0.25 (0.02)    |
> | TACO            | 0.58 (0.05)     | 0.46 (0.04)    | 0.39 (0.01)    | 0.16 (0.03) |
> | AdaLead            | 0.58 (0.05)     | 0.46 (0.04)    | 0.39 (0.01)    | 0.16 (0.03) |
> | Pex            | 0.58 (0.05)     | 0.46 (0.04)    | 0.39 (0.01)    | 0.16 (0.03) |
>
> Table 2
> | **Method**           | **SK-N-SH ↑**     | **HepG2 ↓**     | **K562 ↓** | **ΔR**        |
> |----------------------|-----------------|----------------|----------------|----------------|
> | regCon               | 0.86 (0.02)     | 0.54 (0.13)    | 0.44 (0.01)    | 0.37 (0.11)    |
> | regCon-DE            | 0.68 (0.02)     | 0.28 (0.04)    | 0.26 (0.02)    | 0.41 (0.02)    |
> | TACO            | 0.54 (0.05)     | 0.42 (0.09)    | 0.42 (0.02)    | 0.12 (0.01) |
> | AdaLead            | 0.58 (0.05)     | 0.46 (0.04)    | 0.39 (0.01)    | 0.16 (0.03) |
> | Pex            | 0.58 (0.05)     | 0.46 (0.04)    | 0.39 (0.01)    | 0.16 (0.03) |
>
> ### **[Q4]**
> Yes, a core difference lies in the advantage estimation method. regCon uses a GRPO-inspired approach based on batch-normalized rewards, eliminating the need for a value network. In contrast, PPO-Lagrangian relies on Generalized Advantage Estimation (GAE), which depends on training a value network.
>
> As noted in Section 4.2 (Lines 246), training a value network from sparse, sequence-level rewards is inherently unstable, since it must infer accurate token-level estimates. regCon avoids this issue by directly using normalized rewards, leading to more stable policy updates and potentially explaining the observed performance gap.
> ### **[Q5]**:
> For each target cell type, we initialize three penalty coefficients (λ): one for each off-target constraint (λ = 0.5 for JURKAT, K562, SK-N-SH, HepG2; 0.7/0.3 for THP1, reflecting the [0,1] range of λ) and one for the TFBS constraint (initialized at 0.0 with an upper bound of 0.1). These values were selected by grid search to balance expressions in different cell types and motif enrichment. Although Lagrangian methods can be sensitive to initializations[8], we observed stable performance across all cell types with the same initializations. As noted in the line 332, we plan to incorporate PID controllers [6] to improve λ adaptation in future work.
> ### **Paper Formatting Concerns**
> We will revise Figure 2 to use more distinguishable colors and marker shapes for improved clarity.
>
> Thank you again for your valuable suggestions, which have helped improve the clarity and contribution of our work. Please let us know if you have any more questions.
>
> [1] Lal, Anusha, et al. "Designing realistic regulatory DNA with autoregressive language models." Genome Research, vol. 34, no. 9, 2024, pp. 1411–1420.
>
> [2] Reddy, A. J., et al. "Strategies for effectively modelling promoter-driven gene expression using transfer learning." bioRxiv, 2024, doi:10.1101/2023.02.01.526753.
>
> [3] Avsec, Žiga, et al. "Effective gene expression prediction from sequence by integrating long-range interactions." Nature Methods, vol. 18, no. 10, 2021, pp. 1196–1203.
>
> [4] Yang, Zhen, et al. "Regulatory DNA sequence design with reinforcement learning." The Thirteenth International Conference on Learning Representations, 2025.
>
> [5] Schulman, John, et al. "High-dimensional continuous control using generalized advantage estimation." arXiv preprint arXiv:1506.02438, 2015.
>
> [6] Stooke, Adam, Joshua Achiam, and Pieter Abbeel. "Responsive safety in reinforcement learning by pid lagrangian methods." International Conference on Machine Learning. PMLR, 2020.
>
> [7] Liu, Enbo, et al. "Pareto set learning for multi-objective reinforcement learning." Proceedings of the AAAI Conference on Artificial Intelligence, vol. 39, no. 18, 2025, pp. 18789–18797.
>
> [8] Ji, Jiaming, et al. "Safety Gymnasium: A unified safe reinforcement learning benchmark." Advances in Neural Information Processing Systems, vol. 36, 2023, pp. 18964–18993.

---

> > ### Comment · Reviewer_saSR · 2025-08-04
> > **Remaining Concerns**
> >
> > Thanks for your effort in the rebuttal.
> >
> > I still have the following concerns:
> >
> > 1. Regarding Table 2 in your response, why do Setting 2's results outperform regCon? From an intuitive perspective, Setting 2 creates an inconsistency between the reward model used for guiding optimization and the one used for evaluation, so why are the results still better? I still believe that serious consideration of the reward model is crucial, as relying solely on reward model scores to judge model effectiveness is highly unreliable for DNA CRE tasks.
> >
> > 2. Regarding W4, can you now provide more details since you have additional response space?
> >
> > 3. Concerning GRPO's model performance improvements, if the core enhancement primarily stems from simply using GRPO, I would question where the paper's main contribution truly lies. In Table 1 of the manuscript, PPO-Lag appears to perform quite poorly, even worse than other single-objective baselines, suggesting that the constrained problem formulation may not be effective. However, in Table 2 of the rebuttal, Lag performs much better than DE. The authors need to more clearly decouple the contributions of different components to the overall improvement.

---

> > > ### Author Response · Authors · 2025-08-04
> > > **Response to Q1 (Comment 1 / 3)**
> > >
> > > ## Q1
> > > Thank you for your thoughtful follow-up. We agree that reward model selection and evaluation need careful consideration. Following prior work in DNA regulatory element (CRE) design [1,2], we adopt the regLM reward model as our primary optimization and evaluation model for regCon, due to its prior validation and practical adoption.
> > >
> > > To avoid any miscommunication, we wish to further clarify our response to W2, which was raised by the reviewer regarding reward hacking. We conducted experiments under three reward model variants:
> > >
> > > - Paired model (regLM): Trained excluding chromosomes 7, 13, 21, and 22.
> > > - Full model: Trained on all available sequences.
> > > - Subset model: Trained on a randomly selected 80% subset of the full dataset.
> > >
> > > In this framework, the full model acts as a trusted oracle, approximating the best available estimate of true regulatory activity. The paired and subset models are proxy signals that may carry modeling biases. We report performance under different optimization–evaluation models below:
> > >
> > > | **Method**           | **Optimized with**  | **Evaluated with**  | **HepG2 ↑**     | **K562 ↓**     | **SK-N-SH ↓** | **ΔR**        |
> > > |----------------------|-----------------|----------------|----------------|----------------|-----------------|----------------|
> > > | regCon               |Paired           | Paired         | 0.77 (0.01)     | 0.36 (0.04)    | 0.31 (0.04)    | 0.44 (0.03)    |
> > > | Setting 1            |Paired           | All            | 0.76 (0.02)     | 0.35 (0.03)    | 0.35 (0.02)    | 0.40 (0.04)    |
> > > | Setting 2            |Subset           | All            | 0.82 (0.04)     | 0.37 (0.08)    | 0.36 (0.02)    | 0.45 (0.01)    |
> > >
> > > >Why do Setting 2's results outperform regCon?
> > >
> > > We clarify that Setting 2 uses a different optimization and evaluation model than regCon. Therefore, its performance cannot be directly compared to regCon. We interpret the results as follows:
> > >
> > > - **regCon vs. Setting 1 (same optimization model, different evaluators):**
> > > If regCon were experiencing reward hacking, we would expect a substantial drop in performance when evaluated using a different model. However, regCon maintains strong performance in Setting 1, suggesting robustness and no evidence of reward hacking.
> > >
> > > - **Setting 1 vs. Setting 2 (different optimization models, same evaluator):**
> > > Setting 2 is designed as a robustness test: by optimizing with a reward model trained on only a subset of the data, we test if the trained model can generalize to the oracle reward. The fact that both approaches yield strong performance under the oracle evaluator show that the policy does not overfit to its training reward model, but captures generalizable patterns across distinct training splits.
> > >
> > > Additionally, we mitigate reward hacking by integrating biological priors through motif regularization and applying KL regularization to prevent the policy from diverging too far from the pretrained backbone, a technique shown by [3] to reduce reward hacking.
> > > >  I still believe that serious consideration of the reward model is crucial, as relying solely on reward model scores to judge model effectiveness is highly unreliable for DNA CRE tasks.
> > >
> > > Overall, these findings demonstrate that regCon generalizes across reward models and that its performance is not an artifact of reward model exploitation. While proxy rewards are inherently imperfect, our results support the stability and robustness of the learned policy. We agree that reward models are only proxies, and ultimately, experimental validation is needed to confirm functional activity. While experimental validation remains the ultimate standard, in the domain of DNA CRE design, it is common practice to use trained reward models as proxies to guide in silico sequence optimization [1–5]. Within this context, our aim is not to replace experimental assays, but to provide a practical in silico filtering strategy for prioritizing candidate sequences for downstream validation.
> > >
> > > [1] Yang, Zhen, et al. *Regulatory DNA Sequence Design with Reinforcement Learning*. The Thirteenth International Conference on Learning Representations, 2025.
> > >
> > > [2] Uehara, Maho, et al. "Reward-Guided Iterative Refinement in Diffusion Models at Test-Time with Applications to Protein and DNA Design." *Proceedings of the 42nd International Conference on Machine Learning*, 2025.
> > >
> > > [3] Laidlaw, Cassidy, Shreyas Singhal, and Anca Dragan. "Correlated Proxies: A New Definition and Improved Mitigation for Reward Hacking." *The Thirteenth International Conference on Learning Representations*, 2025.
> > >
> > > [4] Linder, Joshua, Nathan Bogard, Alexander B. Rosenberg, and Georg Seelig. "A Generative Neural Network for Maximizing Fitness and Diversity of Synthetic DNA and Protein Sequences." *Cell Systems*, vol. 11, no. 1, 2020, pp. 49–62.
> > >
> > > [5] Gosai, Shreyas J., et al. "Machine-Guided Design of Cell-Type-Targeting Cis-Regulatory Elements." *Nature*, 23 Oct. 2024, pp. 1–10.

---

> > > > ### Author Response · Authors · 2025-08-05
> > > > **Response to Q2 ( Comment 2/3)**
> > > >
> > > > Thank you for the feedback. We have expanded the baseline descriptions to include implementation details that were previously omitted. Specifically:
> > > >
> > > > AdaLead [1] is implemented as a novelty-guided hill-climbing algorithm with mutation rate μ = 1 (applied as μ/L per position), recombination rate r = 0.2, and greedy threshold κ = 0.05 for selecting parents with fitness above (1–κ) of the current best. Recombination is disabled (ρ = 0), and each candidate is evaluated with v = 20 model queries.
> > > >
> > > > Bayesian Optimization (BO) [1] uses a Gaussian process surrogate and selects candidates via upper confidence bound. We use action sampling, and reset the model state when uncertainty exceeds 1.2× the initial level.
> > > >
> > > > CMA-ES [2] is applied to relaxed one-hot sequence encodings, with a population size of 16 and initial search variance of 0.2. The number of iterations is scaled to the model query budget.
> > > >
> > > > PEX [3] generates candidates by applying 2 random mutations to high-fitness frontier sequences, stratified by Hamming distance from the wild-type. The frontier neighbor size is set to 5, and predictions are processed in batches of 64.
> > > >
> > > > RegLM [4] is an autoregressive language model trained to generate cis-regulatory elements (CREs) conditioned on cell-type-specific fitness profiles. We fine‑tune the hyenadna‑medium‑160k‑seqlen configuration (6.55 M parameters) for 16 epoch, using the AdamW optimizer (learning rate = 0.0001, batch size = 1024) and cross‑entropy loss, with inverse‑frequency label sampling to address class imbalance.
> > > >
> > > > TACO's [5] implementation follows the original paper’s hyperparameter settings. The TFBS reward coefficient is set to 0.01. We use a batch size of 256 and run optimization for 100 iterations. The learning rate is fixed at 1e-4 across all datasets.
> > > >
> > > > For PPO [6], we use Generalized Advantage Estimation (GAE) to compute the advantage function. Optimization is performed using the Adam optimizer with a learning rate of 1e-4. The value network shares the same pretrained HyenaDNA backbone as the policy network, with a linear head on top of the final hidden state to predict value estimates. We use a batch size of 256 and train for 100 iterations.
> > > >
> > > > PPO-Lag [7] extends PPO by introducing a Lagrangian-based constraint formulation. For each reward component (i.e., each cell type), a separate value network is trained to estimate the expected return for that component. These value networks share the same architecture as in PPO. However, due to GPU memory constraints, we apply LoRA [8] to fine-tune only the MLP and mixer layers in the HyenaDNA model. The optimizer, learning rate (1e-4), batch size (256), and total training iterations (100) are kept consistent with the PPO setup.
> > > >
> > > > [1] Sam Sinai, Richard Wang, Alexander Whatley, Stewart Slocum, Elina Locane, and Eric D Kelsic. Adalead: A simple and robust adaptive greedy search algorithm for sequence design. arXiv preprint arXiv:2010.02141, 2020.
> > > >
> > > > [2] Nikolaus Hansen. The cma evolution strategy: a comparing review. Towards a new evolutionary computation: Advances in the estimation of distribution algorithms, pages 75–102, 2006.
> > > >
> > > > [3] Zhizhou Ren, Jiahan Li, Fan Ding, Yuan Zhou, Jianzhu Ma, and Jian Peng. Proximal exploration for model guided protein sequence design. In International Conference on Machine Learning, pages 18520–18536.
> > > >
> > > > [4] Avantika Lal, David Garfield, Tommaso Biancalani, and Gokcen Eraslan. Designing realistic regulatory
> > > > dna with autoregressive language models. Genome Research, 34(9):1411–1420, 2024.
> > > >
> > > > [5] Zhao Yang, Bing Su, Chuan Cao, and Ji-Rong Wen. Regulatory dna sequence design with reinforcement
> > > > learning. In The Thirteenth International Conference on Learning Representations. PMLR, 2022.
> > > >
> > > > [6] John Schulman, Filip Wolski, Prafulla Dhariwal, Alec Radford, and Oleg Klimov. Proximal policy
> > > > optimization algorithms. arXiv preprint arXiv:1707.06347, 2017.
> > > >
> > > > [7]Alex Ray, Joshua Achiam, and Dario Amodei. Benchmarking safe exploration in deep reinforcement
> > > > learning. arXiv preprint arXiv:1910.01708, 7(1):2, 2019.
> > > >
> > > > [8] Hu, Edward J., et al. "Lora: Low-rank adaptation of large language models. arXiv 2021." arXiv preprint arXiv:2106.09685 10 (2021).

---

> > > > > ### Author Response · Authors · 2025-08-05
> > > > > **Response to Q3 ( Comment 3/3)**
> > > > >
> > > > > We appreciate the reviewer's request for clarification regarding our contributions. Our work delivers two separable advances that were both necessary for the strong empirical gains: a constrained problem formulation for multi-objective sequence optimization, and a batch-wise GRPO training approach that enables practical implementation of this formulation.
> > > > >
> > > > > > The authors need to more clearly decouple the contributions of different components to the overall improvement.
> > > > >
> > > > >
> > > > > To address concerns about decoupling these contributions, we conducted one additional experiemnt which is optimizing single reward objective with our batch-wise GRPO. As shown in the table below, we compared regCon (our full method with constrained formulation and batch-wise GRPO), regCon-DE (our batch-wise GRPO framework but using differential expression as the reward objective without constraint formulation), and GRPO (batch-wise GRPO with single reward objective and no constraint formulation).
> > > > >
> > > > >
> > > > > | **Method**           | **HepG2 ↑**     | **K562 ↓**     | **SK-N-SH ↓** | **ΔR**        |
> > > > > |----------------------|-----------------|----------------|----------------|----------------|
> > > > > | regCon               | 0.77 (0.01)     | 0.36 (0.04)    | 0.31 (0.04)    | 0.44 (0.03)    |
> > > > > | regCon-DE            | 0.67 (0.02)     | 0.47 (0.02)    | 0.37 (0.03)    | 0.25 (0.02)    |
> > > > > | GRPO            | 0.76 (0.01)     | 0.84 (0.01)    | 0.82 (0.01)    | -0.05 (0.02) |
> > > > >
> > > > > The constrained problem formulation represents our primary theoretical contribution. Our full method (regCon) achieves high on-target performance (HepG2: 0.77) while maintaining low off-target effects (K562: 0.36, SK-N-SH: 0.31), resulting in ΔR = 0.44. In contrast, regCon-DE (unconstrained with DE objective) shows lower on-target performance (0.67) with higher off-target effects, achieving only ΔR = 0.25. Most critically, GRPO with single objective optimization achieves high on-target performance (0.76) but exhibits catastrophically high off-target effects (0.84, 0.82), resulting in negative ΔR = -0.05. This clearly demonstrates that without the constrained formulation, methods either sacrifice on-target performance or exhibit unacceptable off-target effects, making our constrained approach essential for balancing competing objectives.
> > > > > > In Table 1 of the manuscript, PPO-Lag appears to perform quite poorly, even worse than other single-objective baselines, suggesting that the constrained problem formulation may not be effective. However, in Table 2 of the rebuttal, Lag performs much better than DE
> > > > >
> > > > > Our second contribution, the batch-wise GRPO training approach, addresses the critical scalability challenge that makes constrained multi-objective optimization practically feasible. PPO-Lag's underperformance in Table 1 of the manuscript stems from computational limitations rather than theoretical inadequacy, as mentioned in the manuscript. PPO-Lag requires separate value networks for each constraint (each cell type), creating extreme memory demands and training instability. With N constraints, PPO-Lag needs N additional networks each matching the policy network size, creating a prohibitive computational burden. Our batch-wise GRPO training approach is therefore not merely an incremental improvement but rather the key enabler that makes constrained multi-objective optimization practically feasible. While PPO-Lag could theoretically achieve comparable performance with unlimited computational resources and perfect training stability, our approach provides a scalable solution that maintains the benefits of constrained optimization under realistic resource constraints.

---

> > > > > > ### Comment · Reviewer_saSR · 2025-08-05
> > > > > >
> > > > > > Thank you for the clear and strong rebuttal. I would like to increase my score.

---

### Official Review · Reviewer_49ng · 2025-07-03

**Clarity:** 3
**Significance:** 4
**Originality:** 3
**Rating:** 5
**Confidence:** 4

**Summary:**

This paper introduces regCon, a constrained generative model for DNA sequence design based on constrained reinforcement learning (RL). It aims to increase the cell-type specificity in DNA sequence generation by maximizing fitness on target cell type and minimizing it on off-target ones. It leverages the Lagrangian techniques to enforce the constraints. They benchmark it on MPRA datasets of enhancers and promoters of various cell types show that the generated sequences have high cell type specificity compared to other models suc as regLM. Overall, regCon is an innovative methods for the task of DNA sequence design.

**Questions:**

- How do you choose the best hyperparameters? If you want to design CREs for a new cell type, how would you choose the hyperparameters?

**Ethical Concerns:**

["NO or VERY MINOR ethics concerns only"]

**Final Justification:**

After reading the rebuttal, my concerns are resolved, and I tend to keep my score.

**Limitations:**

Yes

**Quality:**

4

**Strengths And Weaknesses:**

Strengths:

- Adding constraints to the RL framework for DNA generation is innovative and significant.

- The paper is written clearly and explains all the details step by step.

- It is the first os such models that explicitly formulates the off-target fitness minimization.

Weaknesses:

- The models seem to have a lot of hyperparameters.

---

> ### Author Rebuttal · Authors · 2025-07-30
>
> We thank the reviewer for the positive feedback and the insightful question.
>
> **How to choose the best hyperparameters**
>
> We performed a grid search over key hyperparameters such as the KL coefficient, learning rate, and multipliers. We select hyperparameters by running experiments across different hyperparameters and monitoring reward-based metrics (e.g., target activity, constraint satisfaction) on the generated sequences.
>
> **How to choose the hyperparameters for a new cell type**
>
> When designing for a new cell type, we use the corresponding reward model to guide optimization. In practice, we initialize with hyperparameters that worked well on similar cell types and find that our method is relatively robust across a wide range of settings. As shown in Table 3 (line 477 in the manuscript and also down below), many cell types share the same hyperparameters, suggesting generalizability across targets.
>
> We acknowledge that our method involves a number of hyperparameters. This is in part due to its foundation in Lagrangian-based reinforcement learning [1]. However, many of these hyperparameters can be reused across cell types. For the reviewer’s convenience, we have included the hyperparameter table from the appendix in the rebuttal below. Users can start from the configurations we provide and refine them through a lightweight grid search based on training dynamics.
>
> **Experiment Hyperparameters**
>
> | Hyperparameter                          | HepG2 | K562 | SK-N-SH | JURKAT | K562 | THP1 |
> |----------------------------------------|:-----:|:----:|:-------:|:------:|:----:|:----:|
> | Batch Size                             |  256  | 256  |   256   |  256   | 256  | 256  |
> | Replay Buffer Batch Size               |   24  |  24  |    24   |   24   |  24  |  24  |
> | Policy Learning Rate ($η_θ$)   | 1e-4  | 1e-4 |  1e-4   |  1e-4  | 1e-4 | 1e-4 |
> | Multiplier Learning Rate ($η_λ$)| 3e-4 | 3e-4 |  3e-4   |  3e-4  | 3e-3 | 3e-3 |
> | KL Value Coefficient (β)              |  0.2  | 0.2  |   0.2   |  0.2   | 0.2  | 0.2  |
> | TFBS Multiplier Upper Bound ($λ_{max}$) | 0.1 | 0.1 |  0.1   |  0.1   | 0.1  | 0.1  |
>
> Thank you again for your constructive review and positive comments. Please let us know if you have any further questions!
>
> **Reference**
>
> [1] Ray, A., Achiam, J., & Amodei, D. (2019). Benchmarking safe exploration in deep reinforcement learning. arXiv preprint arXiv:1910.01708, 7(1), 2.

---

> > ### Comment · Area_Chair_2Wi5 · 2025-08-05
> > **Reminder to respond to the rebuttal**
> >
> > Dear Reviewer 49ng,
> >
> > Thank you again for reviewing this paper. Since the author-reviewer discussion phase is closing soon, could you please respond to the authors' rebuttal?
> >
> > Best,
> >
> > AC

---

> > ### Comment · Reviewer_49ng · 2025-08-06
> >
> > I would like to thank the authors for addressing my questions. All my concerns have been resolved. I tend to keep my score.

---

### Official Review · Reviewer_TfWB · 2025-07-03

**Clarity:** 4
**Significance:** 4
**Originality:** 4
**Rating:** 5
**Confidence:** 4

**Summary:**

Designing cell-type-specific cis-regulatory elements (CREs) such as promoters and enhancers are vital to the success of gene therapies. This work proposes a constrained RL-based method called regCon to design such CREs. regCon fine-tunes an autoregressive genomic language model using constrained RL to maximize reward model-predicted expression in a target cell type while maintaining predicted expression in non-targeted cells below a specified threshold. regCon uses Lagrange multipliers to control cell-type-specificity and policy gradients for optimizing the policy. Instead of adopting the conventional approach of using learned value functions, regCon instead modifies the Group Relative Policy Optimization (GRPO) framework to use batch-level information to compute advantage estimates – this incentivizes the model to produce sequences that have rewards similar to the best sequences in the batch. Finally, to ensure that the generated sequences retain transcription factor (TF) binding motifs observed in real cell-type-specific sequences (and thus reduce reward hacking), the authors add a constraint reward function that incentivizes the model to retain these motifs. The authors thoroughly benchmark regCon using two datasets that measure CRE activity and compare its performance to many baselines. regCon outperforms all baselines convincingly, produces cell-type-specific TF-binding motifs, and further ablation analyses validate the usefulness of various components.

**Questions:**

I have mentioned my questions and suggestions in the previous section.

**Ethical Concerns:**

["NO or VERY MINOR ethics concerns only"]

**Final Justification:**

This submission was already very strong and the authors further improved it during the discussion period by providing more baselines. I have no other issues with the paper, and retain my original recommendation to accept it.

**Limitations:**

Yes.

**Quality:**

4

**Strengths And Weaknesses:**

This is a strong submission in my opinion with well-motivated techniques, thorough benchmarking, and clear writing. I provide more details below.

Quality:
- The paper is technically sound and all parts of the regCon framework are clearly motivated and build on relevant prior work
- Claims made in the paper are convincingly supported by the results - it is clear from the analyses that regCon is a potent sequence optimizer for designing cell-type-specific CREs since it clearly outperforms baselines.
- Have the authors tried using a relatively simple reward function such as the average difference in expression between the target and non-target cells (like in Reddy et al. [1]) instead of using the constraint-based reward function? Adding this result would make the case for using the constraint-based reward function more convincing.
- Although the authors benchmark many different baselines, they do not benchmark prior biological sequence design methods such as those proposed by Reddy et al. [1], Linder et al. [2], Brookes et al. [3], Gosai et al [4], etc. Adding some of these as additional baselines might make the case for regCon even more convincing.

Clarity: In general, this paper is very well-written and easy to understand. My only suggestion is to have more details about how exactly the datasets are processed. The authors currently reference another paper and say they use the same pipeline but it would be useful to explicitly write out the processing steps in the appendix for the sake of reproducibility.

Significance:
- In my opinion, this could be a high impact submission given that it builds on prior work to provide a sequence optimization framework that could be of broad use to computational biologists. It also aims to solve a difficult yet important problem.
This work would also be of interest to RL researchers given its innovations
- Apart from cell-type-specific CRE design, this framework could be adapted for solving other constraint-driven optimization problems that are common in biology such as protein or RNA design.

Originality: I am quite familiar with biological sequence design techniques but I am not as familiar with the latest RL research. However, to the best of my knowledge, this paper presents many original ideas such as using the constrained RL framework for cell-type-specific CRE design, using batch-level GRPO, and incorporating TF-binding motif retention into the optimization process. As mentioned previously, the authors provide clear motivations for these ideas and how they build on prior work.

References:
1. Reddy AJ, Geng X, Herschl M, Kolli S, Kumar A, Hsu P, Levine S, Ioannidis N. Designing cell-type-specific promoter sequences using conservative model-based optimization. Advances in Neural Information Processing Systems. 2024 Dec 16;37:93033-59.
2. Linder J, Bogard N, Rosenberg AB, Seelig G. A generative neural network for maximizing fitness and diversity of synthetic DNA and protein sequences. Cell systems. 2020 Jul 22;11(1):49-62.
3. Brookes D, Park H, Listgarten J. Conditioning by adaptive sampling for robust design. InInternational conference on machine learning 2019 May 24 (pp. 773-782). PMLR.
4. Gosai SJ, Castro RI, Fuentes N, Butts JC, Mouri K, Alasoadura M, Kales S, Nguyen TT, Noche RR, Rao AS, Joy MT. Machine-guided design of cell-type-targeting cis-regulatory elements. Nature. 2024 Oct 23:1-0.

---

> ### Author Rebuttal · Authors · 2025-07-30
>
> We appreciate the reviewer’s recognition of our method’s novelty, the concreteness of our experiments, and the framework’s potential for broader biological design tasks. We also thank the reviewer for the constructive feedback. Please find our reponse below regarding the added baselines experiemnts and clarification on dataset processing.
>
>
> ###  **Comparison with simple DE-based reward functions and other baselines**
> **Summary**:
>
> We added the requested baselines and differential expression (DE) as a scalar reward. Across enhancer and promoter settings, regCon yields higher on target expression while meeting off target constraints. Scalar DE optimization underperforms and often increases DE by suppressing off target at the cost of on target signal.
>
> **Detail response:**
>
> We have added experiments involving the baselines mentioned: Reddy et al. [1], Linder et al. [2], Brookes et al. [3], and Gosai et al. [4], as well as additional experiments using DE as a scalar reward function (denoted with a "-DE" suffix in Table 1-3).
>
> Specifically, Gosai et al. [4] proposed three generation strategies: AdaLead, Fast SeqProp (FSP), and Simulated Annealing (SA). Since AdaLead was already included in our main experiments, we additionally implemented and benchmarked FSP and SA. Due to time and resource constraints, we evaluated:
>
> - Brookes et al. [3] and Gosai et al. [4] (FSP and SA) on the human enhancer dataset, alongside the DE-based variants of our baselines (Table 1-3).
> - Reddy et al. [1] and Linder et al. [2] on the human promoter dataset (Table 4-6).
>
> As shown across all tables, DE-based optimization consistently underperforms compared to regCon. Scalar DE optimization applies a fixed linear trade-off (Target - Off), which tend to converge to suboptimal optimization space[5]. In contrast, Lagrangian methods dynamically adjust penalty coefficients through learned multipliers, enabling the model to escape poor local optima and achieve more effective optimization.
>
> Furthermore, maximizing DE as a scalar reward can lead the model to aggressively suppress off-target expression, even at the expense of reducing on-target activation. This behavior is evident in the performance of regCon-DE in Table 3, where the method achieves higher DE but at the expense of significantly lower on-target expression compared to regCon. In contrast, regCon enforces explicit off-target constraints and directs optimization toward increasing on-target activity within the feasible region, leading to higher on-target cell activity and more biologically meaningful solutions.
>
>
>
> **Table 1.**
> | **Method**           | **HepG2 ↑**     | **K562 ↓**     | **SK-N-SH ↓** | **ΔR**        |
> |----------------------|-----------------|----------------|----------------|----------------|
> | regCon               | 0.77 (0.01)     | 0.36 (0.04)    | 0.31 (0.04)    | 0.44 (0.03)    |
> | regCon-DE            | 0.67 (0.02)     | 0.47 (0.02)    | 0.37 (0.03)    | 0.25 (0.02)    |
> | TACO-DE              | 0.58 (0.05)     | 0.46 (0.04)    | 0.39 (0.01)    | 0.16 (0.03)    |
> | AdaLead-DE           | 0.45 (0.03)     | 0.46 (0.05)    | 0.43 (0.13)    | -0.01 (0.02)    |
> | Pex-DE               | 0.67 (0.06)     | 0.57 (0.03)    | 0.61 (0.04)    | 0.06 (0.03)    |
> | Brookes et al. [3]   | 0.74 (0.03)     | 0.79 (0.04)    | 0.73 (0.03)    | -0.02 (0.02)  |
> | FSP [4]                  | 0.62 (0.03)     | 0.36 (0.02)    | 0.36 (0.04)    | 0.26 (0.03)    |
> | SA  [4]                 | 0.61 (0.01)     | 0.33 (0.01)    | 0.33 (0.01)    | 0.28 (0.03)    |
>
> **Table 2.**
> | **Method**           | **K562 ↑**     | **HepG2 ↓**     | **SK-N-SH ↓** | **ΔR**        |
> |----------------------|-----------------|----------------|----------------|----------------|
> | regCon               | 0.93 (0.01)     | 0.43 (0.01)    | 0.35 (0.01)    | 0.54 (0.01)    |
> | regCon-DE            | 0.87 (0.04)     | 0.43 (0.01)    | 0.35 (0.02)    | 0.48 (0.03)    |
> | TACO-DE              | 0.80 (0.01)     | 0.48 (0.01)    | 0.49 (0.02)    | 0.50 (0.01) |
> | AdaLead-DE            | 0.54 (0.16)     | 0.49 (0.12)    | 0.49 (0.15)    | 0.05 (0.03) |
> | Pex-DE               | 0.81 (0.04)     | 0.60 (0.06)    | 0.61 (0.08)    | 0.18 (0.07) |
> | Brookes et al. [3]   | 0.90 (0.01)     | 0.66 (0.02)    | 0.69 (0.01)    | 0.23 (0.18) |
> | FSP [4]                  | 0.67 (0.07)     | 0.35 (0.02)    | 0.32 (0.04)    | 0.33 (0.04) |
> | SA [4]            | 0.67 (0.07)     | 0.34 (0.01)    | 0.33 (0.01)    | 0.33 (0.04) |
>
> **Table 3.**
> | **Method**           | **SK-N-SH ↑**     | **HepG2 ↓**     | **K562 ↓** | **ΔR**        |
> |----------------------|-----------------|----------------|----------------|----------------|
> | regCon               | 0.86 (0.02)     | 0.54 (0.13)    | 0.44 (0.01)    | 0.37 (0.11)    |
> | regCon-DE            | 0.68 (0.02)     | 0.26 (0.02)    | 0.28 (0.04)    | 0.41 (0.02)    |
> | TACO-DE              | 0.54 (0.02)     | 0.42 (0.04)    | 0.42 (0.09)    | 0.12 (0.01) |
> | AdaLead-DE            | 0.45 (0.03)     | 0.43 (0.03)    | 0.45 (0.05)    | 0.01 (0.02) |
> | Pex-DE               | 0.77 (0.03)     | 0.64 (0.09)    | 0.67 (0.09)    | 0.09 (0.02) |
> | Brookes et al. [3]   | 0.81 (0.05)     | 0.68 (0.03)    | 0.81 (0.06)    | 0.06 (0.01) |
> | FSP [4]          | 0.54 (0.03)     | 0.39 (0.02)    | 0.36 (0.01)    | 0.16 (0.02) |
> | SA [4]            | 0.54 (0.04)     | 0.37 (0.05)    | 0.35 (0.04)    | 0.18 (0.05) |
>
> **Table 4.**
> | **Method**           | **JURKAT ↑**     | **K562 ↓**     | **THP1 ↓** | **ΔR**        |
> |----------------------|-----------------|----------------|----------------|----------------|
> | regCon               | 0.69 (0.09)     | 0.38 (0.02)    | 0.49 (0.01)    | 0.25 (0.01)    |
> | Reddy et al. [1]     | 0.47 (0.02)      | 0.31 (0.01)   | 0.51 (0.01)    | 0.06 (0.01)    |
> | Linder et al. [2]    | 0.40 (0.05)     | 0.25 (0.03)    | 0.51 (0.01)    | 0.03 (0.05)    |
>
>
> **Table 5.**
> | **Method**           | **K562 ↑**     | **JURKAT ↓**     | **THP1 ↓** | **ΔR**        |
> |----------------------|-----------------|----------------|----------------|----------------|
> | regCon               | 0.58 (0.03)     | 0.43 (0.06)    | 0.50 (0.01)    | 0.12 (0.02)    |
> | Reddy et al. [1]     | 0.41 (0.01)      | 0.48 (0.02)   | 0.75 (0.01)    | -0.12 (0.01)   |
> | Linder et al. [2]    | 0.24 (0.03)     | 0.39 (0.03)    | 0.51 (0.01)   | -0.21 (0.05)    |
>
> **Table 6.**
> | **Method**           | **THP1 ↑**     | **JURKAT ↓**     | **K562 ↓** | **ΔR**        |
> |----------------------|-----------------|----------------|----------------|----------------|
> | regCon               |0.92 (0.01)     | 0.51 (0.01)    | 0.31 (0.03)    | 0.51 (0.01)    |
> | Reddy et al. [1]     | 0.55 (0.02)    | 0.39 (0.01)    | 0.29 (0.01)    | 0.20 (0.01)   |
> | Linder et al. [2]    | 0.51 (0.01)     |0.41 (0.07)    | 0.22 (0.03)   | 0.20 (0.03)    |
>
> ### **Clarity: add dataset processing details**
> Thank you for the suggestion. We will add an Appendix section that details data processing steps to enhance reproducibility.
>
> Thank you again for your helpful feedback. We’re happy to address any further suggestions or questions.
> ### **References:**
> [1] Reddy AJ, Geng X, Herschl M, Kolli S, Kumar A, Hsu P, Levine S, Ioannidis N. Designing cell-type-specific promoter sequences using conservative model-based optimization. Advances in Neural Information Processing Systems. 2024 Dec 16;37:93033-59.
>
> [2] Linder J, Bogard N, Rosenberg AB, Seelig G. A generative neural network for maximizing fitness and diversity of synthetic DNA and protein sequences. Cell systems. 2020 Jul 22;11(1):49-62.
>
> [3] Brookes D, Park H, Listgarten J. Conditioning by adaptive sampling for robust design. InInternational conference on machine learning 2019 May 24 (pp. 773-782). PMLR.
>
> [4] Gosai SJ, Castro RI, Fuentes N, Butts JC, Mouri K, Alasoadura M, Kales S, Nguyen TT, Noche RR, Rao AS, Joy MT. Machine-guided design of cell-type-targeting cis-regulatory elements. Nature. 2024 Oct 23:1-0.
>
> [5] Liu, E., Wu, Y. C., Huang, X., Gao, C., Wang, R. J., Xue, K., & Qian, C. (2025, April). Pareto set learning for multi-objective reinforcement learning. In Proceedings of the AAAI Conference on Artificial Intelligence (Vol. 39, No. 18, pp. 18789-18797).

---

> > ### Comment · Area_Chair_2Wi5 · 2025-08-05
> > **Reminder to respond to the rebuttal**
> >
> > Dear Reviewer TfWB,
> >
> > Thank you again for reviewing this paper. Since the author-reviewer discussion phase is closing soon, could you please respond to the authors' rebuttal?
> >
> > Best,
> >
> > AC

---

> > ### Comment · Reviewer_TfWB · 2025-08-05
> >
> > Thank you for the detailed response. All of my questions have been answered and I will keep my score. It would be great if the authors could also detail how these benchmarks were run in the final version of the paper, possibly in the appendix.

---

> > > ### Author Response · Authors · 2025-08-05
> > >
> > > We appreciate your thoughtful feedback and the time you took to review our work. We’re glad to hear that your concerns have been addressed. We will include additional details on the benchmarks in the final version of the paper. Thank you again for your helpful suggestions.

---

### Note · Authors · 2025-08-13

We sincerely thank all reviewers and the Area Chair for their thoughtful feedback, constructive engagement, and recognition of our work. Throughout the discussion period, we provided clarifications, additional analyses, and new experimental results to address the main concerns raised.


- **Comparisons with differential expression rewards and role of constrained optimization:** In response to reviewer suggestions, we compared regCon’s constraint-based reward with differential expression–based rewards and found that the constrained formulation consistently delivers higher target expression alongside greater specificity, outperforming all prior methods. These results highlight that framing cis-regulatory elements (CRE) design as a constrained optimization problem is particularly effective, and baselines with differential expression rewards underperform because of limited adaptive control. We further clarified regCon’s distinction from PPO-Lagrangian, attributing its advantage to stronger constraint formulation and adaptive handling.

- **Additional Baselines and training details**: We included results from additional prior biological sequence design methods, further confirming regCon’s superior performance in optimizing CRE sequence activity and target specificity, and clarified our hyperparameter selection process along with added details on reward model training and data partitioning.

- **Robustness to reward hacking:** To address concerns about overfitting to a particular reward model, we conducted experiments in which training and evaluation used different reward models. The results show consistent performance, indicating that regCon’s gains are not attributable to reward hacking.

- **Motif Correlation Clarification:** We detailed our computation method, justified the percentile threshold, and addressed potential circularity by re-evaluating with distinct motif reference sets, where regCon maintained strong correlations and outperformed all baselines despite reduced data.

We believe these clarifications and additional results address the reviewers’ concerns and underscore the key contributions of our work: introducing a principled, constraint-based framework for CRE design, achieving superior target specificity with greater controllability, and demonstrating robust performance across diverse settings. We sincerely thank the reviewers and the Area Chair for helping strengthen both the clarity and rigor of our paper.

---

### Decision · Program_Chairs · 2025-09-17

**Decision:**

Accept (spotlight)

**Comment:**

This paper presents regCon, which models the cell-type-specific regulatory DNA design problem as a constrained optimization problem and utilizes the GRPO algorithm as the RL framework for sequence optimization. Through evaluations on human promoter and enhancer benchmarks, regCon consistently outperforms existing baselines.

Initially, reviewers maintained a consistently positive attitude toward this work, but several major concerns remained: 1) insufficient details in experimental descriptions, e.g., dataset partitioning and baseline selection; 2) potential reward hacking issues; 3) how to separately evaluate the benefits brought by the constrained optimization problem formulation and batch-level GRPO, respectively; 4) missing diffusion model baselines. After the rebuttal, reviewers acknowledged that these issues have been addressed. The AC thus believes this paper is now ready for acceptance. The camera-ready version should include the discussions and results provided in the rebuttal.